# Pooled PPIseq: Screening the SARS-CoV-2 and human interface with a scalable multiplexed protein-protein interaction assay platform

Darach Miller[1]*, Adam Dziulko[1,2,3], Sasha Levy[1¤]*

1 SLAC National Accelerator Laboratory, Stanford University, Stanford, California, United States of America, 2 Department of Molecular, Cellular, and Developmental Biology, University of Colorado Boulder, Boulder, Colorado, United States of America, 3 BioFrontiers Institute, University of Colorado Boulder, Boulder, Colorado, United States of America

¤ Current address: BacStitch DNA Inc, South San Francisco, California, United States of America
* darach@rhesis.com (DM); sasha@bacstitchdna.com (SL)

## Abstract

Protein-Protein Interactions (PPIs) are a key interface between virus and host, and these interactions are important to both viral reprogramming of the host and to host restriction of viral infection. In particular, viral-host PPI networks can be used to further our understanding of the molecular mechanisms of tissue specificity, host range, and virulence. At higher scales, viral-host PPI screening could also be used to screen for small-molecule antivirals that interfere with essential viral-host interactions, or to explore how the PPI networks between interacting viral and host genomes co-evolve. Current high-throughput PPI assays have screened entire viral-host PPI networks. However, these studies are time consuming, often require specialized equipment, and are difficult to further scale. Here, we develop methods that make larger-scale viral-host PPI screening more accessible. This approach combines the mDHFR split-tag reporter with the iSeq2 interaction-barcoding system to permit massively-multiplexed PPI quantification by simple pooled engineering of barcoded constructs, integration of these constructs into budding yeast, and fitness measurements by pooled cell competitions and barcode-sequencing. We applied this method to screen for PPIs between SARS-CoV-2 proteins and human proteins, screening in triplicate >180,000 ORF-ORF combinations represented by >1,000,000 barcoded lineages. Our results complement previous screens by identifying 74 putative PPIs, including interactions between ORF7A with the taste receptors TAS2R41 and TAS2R7, and between NSP4 with the transmembrane KDELR2 and KDELR3. We show that this PPI screening method is highly scalable, enabling larger studies aimed at generating a broad understanding of how viral effector proteins converge on cellular targets to effect replication.

## Introduction

Emerging and endemic viral pathogens are a persistent threat to human health. These obligate parasites infect susceptible cells and redirect human cellular processes towards the replication

**Data Availability Statement:** Please see the "Data Access" section of the manuscript, copied here: "Sequencing datasets are available on the Sequence Read Archive. For long-read sequencing

(PacBio and Nanopore) used to annotate DNA barcodes with the linked ORF they represent, see PRJNA1073210. For Illumina sequencing of the double-barcode amplicon used to quantify the fitness of each diploid lineage, see PRJNA1073201. See the supplemental table (S13) for the more metadata to contextualize these. For other intermediate files see the supplemental files for this paper, or retrieve them as a sqlite3 database file from this OSF repository (doi.org/10.17605/OSF. IO/B8G3H). For example, because of the particularly large size of the counts table, that table is available as a separate sqlite3 database in the linked OSF repository. Raw TIFF scans of the agar plates used for validation of the modified mDHFR assay (Fig 1) or PPI re-testing on agar media (analyzed for Fig 3), is available in the above OSF repository in an appropriate folder."

**Funding:** This work was solely funded by the National Institutes of Health National Institute for Allergies and Infections Diseases (https://www. niaid.nih.gov/) grant R01 AI164530 awarded to Dr. Sasha Levy (SL) and carried out within Dr. Sasha Levy's research group at Stanford University. The funders had no role in study design, data collection and analysis, decision to publish, or preparation of the manuscript.

**Competing interests:** The authors have declared that no competing interests exist.

and release of transmissible viral particles. Studying how this process functions at the molecular scale is important for understanding the mechanisms of pathogenesis and the determinants of cell-type susceptibility and permissibility. **P**rotein-**P**rotein **I**nteractions (**PPIs**) are an important mechanistic factor in a viral infection, with PPIs contributing to both viral replication and host restriction of viral replication [1–4]. Thus viral-human PPIs (**vhPPIs**) are a complex and critical interface between virus and host where genetic variation can partly determine viral host range and virulence [5–9]. Mapping these vhPPIs will establish systematic insight into how interacting viral proteins converge on common human protein targets, cellular functions, or pathways, and identify interactions leading to therapeutic strategies [10].

A variety of techniques have been used to assay PPIs at proteome-scale. Mass-spectrometry after affinity-purification [11–13] or proximity-labeling [14] methods are perhaps most easily scalable to screening virus-host PPIs as they only require engineering a tag fusion to the "bait" viral open reading frame (ORF), but these approaches are also less sensitive to transient PPIs that may be common for regulatory interactions [15–19] or to PPIs involving membrane-associated proteins critical for the complex morphogenesis of viral replication centers or viral particles [20]. Split-reporter constructs assayed in yeast, canonically exemplified by the yeast two-hybrid assay [21–23], are another class of approaches that fuse each "tag" of a split reporter gene to a protein pair of interest such that interaction of the chosen proteins reconstitutes the reporter's function. One such split-reporter assay is the murine dihydrofolate reductase (mDHFR) Protein-fragment Complementation Assay (PCA) [24, 25], and although the proteins of interest are assayed in a transgenic context of budding yeast, the mDHFR PCA has the advantages of being sensitive to membrane-bound proteins, reporting on transient interactions, and not requiring that proteins of interest be localized to the nucleus while not interfering with the assay's reporter gene transcription. Yeast-based split-reporters have been quantified in the past with colony growth [26–28], but can also leverage barcode-sequencing to multiplex PPI assays within a single pooled culture and sequencing library [29–33]. Each method for screening PPI networks introduces diverse biases in detection, therefore, multiple complementary approaches are often compared to generate a more thorough functional and biochemical understanding [17, 34]. Machine-learning prediction of protein 3D structures [35, 36] has also recently been used to predict PPIs [37]. However, these models currently have limited capabilities to generalize to dynamic and disordered protein ensembles (~30% of the human proteome, [38]). These dynamic and disordered vhPPIs are especially important for understanding viral cell biology [8, 39–43], suggesting that empirical PPI screens remain important in exploring cell biology.

A challenge for scaling any approach that relies on a synthetic tag-fusion is the engineering of these DNA constructs. One advance that accelerates high-throughput construct engineering is the Gateway system [44], which was developed to easily move ORF sequences from "Entry" plasmids into "Destination" plasmids that bear the appropriate expression and assay systems. Efforts like the ORFeome Collaboration have generated a collection of Entry plasmids bearing **ORF**s that encode **h**uman proteins (**hORFs**) [45], and this collection can be easily integrated into Destination plasmids for multiple PPI screening approaches. Thus the ORFeome Collection has been key to mapping PPI networks at proteome-scale [46, 47], and efforts to generate **v**iral **ORF** (**vORF**) collections also offer the opportunity to expand our systematic understanding of pan-viral-human PPI networks [48–50]. Current PPI assay platforms require generating or isolating each assay construct as clones. While methods have been developed to perform this cloning in high throughput [33], the overhead and resources required limits proteome-scale PPI network screening to a handful of well-resourced and specialized research groups. Modern pooled DNA engineering methods can be leveraged to integrate (or synthesize *de novo*) ORFs from more viruses and natural genetic variants. When combined with highly

scalable barcode-based PPI assay quantification, pooled DNA engineering may permit a wider range of research groups to address research questions at unprecedented scales.

We previously reported an approach called PPIseq [29, 51] that combined the yeast mDHFR split-tag PPI assay collection [28] with our double-barcode iSeq2 approach [52] to screen ~9% of the potential yeast PPI network in 9 growth conditions. The mDHFR assay works by rescuing the growth defect of yeast upon methotrexate (**MTX**) treatment, and the mDHFR split-tags **F[1,2]** and **F[3]** can be fused to target proteins to assay the target protein's proximity as a reconstitution of the full mDHFR and thus a quantitative rescue of growth [24, 28, 53–56]. We were able to leverage an existing collection of yeast strains, with mDHFR split-tags fused at the native protein locus [28], by introducing our iSeq2 barcoding system by mating and selection. This permits us to use the Cre/lox system and an artificial-intron split-marker selection to generate recombinants where previously known barcodes can be cheaply quantified in pools by amplicon short-read Illumina sequencing. However, this approach required that tagged-protein haploid strains are generated and maintained in clonal arrays where each position's genotype is previously known. This approach bears both the assay tagged-protein construct and the double-barcode locus on the chromosome, and, although these may have advantages over assay constructs expressed from an episome [57, 58], chromosomally-integrated reporters and barcodes are more challenging to engineer and assay with amplicon sequencing. Plasmid-based engineering and quantification has proven successful for various reporters in yeast expression, including the mDHFR split-tag [33], and we sought to offer an alternative strategy of pooled library generation and assay using the iSeq2 interaction-barcoding system.

Here, we expand the utility of the PPIseq system by leveraging pooled engineering techniques to screen for PPIs amongst libraries of heterologous ORFs from Gateway ORFeome plasmid pools. We develop and demonstrate this approach by generating and annotating a pool of mDHFR-tagged and DNA-barcoded ORF expression plasmids bearing ORFs encoding human or SARS-CoV-2 proteins. We then assayed this PPIseq strain collection to screen for PPIs between ~9,000 human ORFs and ~30 SARS-CoV-2 ORFs in biological triplicate. We found that this method largely agrees with, but is, as expected, complementary to other PPI screening approaches, and finds potential novel interactions between proteins of SARS-CoV-2 and human proteins involved in, for example, ER retention and taste receptors.

## Results

### Scaling PPIseq to efficiently screen new ORF libraries

A major bottleneck to PPI screening assays is the engineering of the DNA constructs to be tested. To overcome this challenge, we aimed to develop a PPI screening platform that utilizes pooled DNA engineering techniques. We generate each mDHFR-tagged ORF expression construct on a plasmid, integrate a random barcode into this plasmid, and use long-read sequencing to annotate an expression construct with the physically-linked barcodes (Fig 1A). We use our iSeq 2.0 system [52] to integrate these plasmids into genomic landing pads of two complementary yeast strains, and then combine them onto the same chromosome via yeast mating and recombination between homologous chromosomes. Once combined on the same chromosome, the barcodes from each plasmid are in close enough proximity to be PCR amplified and sequenced on a paired-end Illumina read (Fig 2A). Because all steps of this process are performed in pools, it is, in theory, highly scalable.

To develop this pooled construction and screening approach, we adapted the iSeq 2.0 pBAR4 and pBAR5 plasmids [52] to express an integrated ORF as an in-frame fusion with the mDHFR F[1,2] or F[3] tags [24, 28, 53, 59] from a *TDH3* promoter. This design uses the

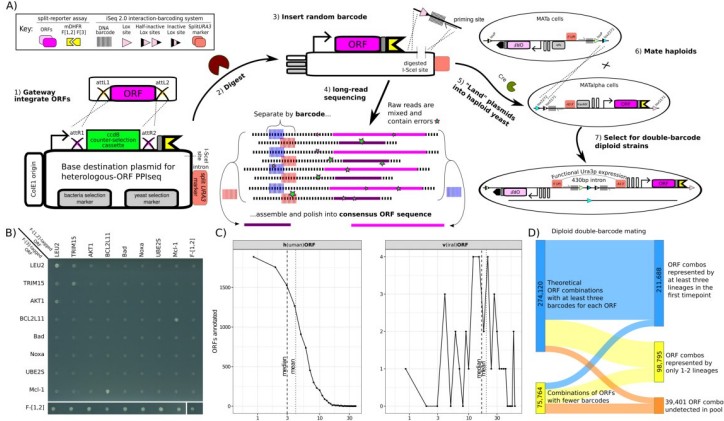

**Fig 1. Complex PPIseq libraries were generated from plasmid pools using a scalable high-throughput workflow.**
**A)** Diagram of the workflow. 1) An *in vitro* recombinase integrates ORFs from an Entry vector pool into Gateway Destination plasmids, 2) the plasmids are digested at a *I-SceI* restriction site, and 3) *in vitro* recombination and cloning repairs plasmid with a lox site, barcode, and priming site, 4) Linearized plasmid pools are sequenced by long-read sequencing to determine the identity of the barcode and ORF integrated in each plasmid, 5) plasmids are "landed" into the yeast iSeq 2.0 genomic "landing pads" using transformation and Cre recombinase, 6) haploid pools of yeast are mated and Cre recombinase recombines the loci to form a double-barcode locus, and 7) the reconstituted split-*URA3* marker is selected for to select the final double-barcoded diploid pool, ready to assay. **B)** A scanned image of double-barcoded diploid strains grown on media selective for the mDHFR reporter (SC-URA+100ug/mL MTX, raw image S1 Fig). ORF combinations are indicated on the margins, with F[1,2]-tagged ORF named along columns and F[3]-tagged ORF named along rows. TRIM15, AKT1, Bim, Bad, Noxa, UBE2S, and Mcl-1 are from the human ORFeome v8.1 collection, LEU2 from the yeast ORFeome collection, and the mDHFR F[1,2] and mDHFR F[3] tagged ORFs were generated here (Methods). **C)** A complex library of barcoded and mDHFR-tagged ORFs can be generated by transformation. The distribution of the number of unique barcoded plasmids detected per ORF, for human and viral ORF plasmid libraries, is plotted as lines and points, with each distribution's mean and median denoted by vertical lines. **D)** A complex library of ORF combinations can be generated by mating. A Sankey diagram illustrates that most (~60%) of the possible (~350k) diploid double-barcode PPI-reporter ORF-ORF combinations that can be generated are observed to be represented by at least three barcode-replicate lineages in the initial library sample.

Gateway system [44] to insert ORFs lacking a stop codon from compatible Gateway Entry ORF libraries [45, 48]. We then developed methods to insert a random DNA barcode ($N_{30}$) from a highly complex pool of oligos after this step. Inserting barcodes by this method minimizes the "collision" of one barcode being associated with multiple ORFs. To enable future studies screening small-molecule perturbations of PPIs (not described here), we modified our base yeast landing-pad strains [52] to delete four ABC transporter genes that are associated with sensitivity of yeast to small-molecules: *PDR1*, *PDR3*, *PDR5*, and *SNQ2* [60]. To enable adequate sampling of chromosomally integrated double-barcodes during amplification from high-complexity barcoded cell libraries, we developed a genomic DNA extraction (S2 File) and library-prep PCR (S3 File) protocol to enable cheap and efficient PCR of double-barcodes from large inputs of quantities of genomic DNA. This amplicon design incorporates a four barcode amplicon design [61, 62] to allow scalable sample multiplexing while bioinformatically detecting chimeras generated by the Illumina sequencing workflow [63, 64].

To determine whether PPIs could still be detected in this modified platform, we constructed and tested the following controls: 1) a positive control mDHFR F[1,2]-F[3] fusion such that the covalent linkage between tags maximally rescues growth of yeast upon MTX addition, 2) a negative control F[1,2]-F[1,2] fusion that is incapable of such a growth rescue, and 3) each negative control mDHFR F[1,2] or F[3] tag fused to only a methionine codon (a NULL construct). These negative control NULL-F[1,2] and NULL-F[3] constructs serve to control for spurious interaction between proteins and the mDHFR tags [34, 65]. We tested the PPI assay (Fig 1B)

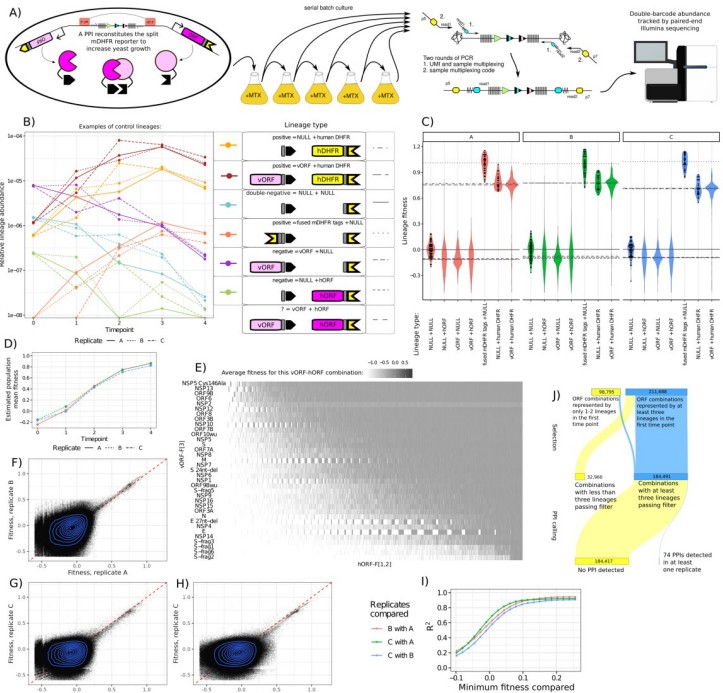

**Fig 2. PPIseq enables reproducible screening of a complex PPI reporter library. A)** Schematic of the assay workflow. **B)** Examples of growth trajectories of each type of control strain generated in this pool, where points and lines denote the relative abundance of a lineage. For each control strain, a randomly selected lineage is shown from each of the three replicate cultures (line style). X-axis denotes the time point when the sample was taken. **C)** Violin plots of the relative fitness for experimental (vORF + hORF) and control (all other) lineages, separated by lineage type. Dot plots of the underlying data are shown for some lineage types. Lineages with low counts and outliers are excluded (Results, Methods). Horizontal line is the median of each lineage type's fitnesses, with linetype corresponding to the lineage type. **D)** The mean fitness of the population as estimated by FitSeq. E) Heatmap of the fitness of each PPI assayed, averaged across all three replicates. Note that NSP2 is adjusted to mitigate bias, see Results. **F,G,H)** Fitness of lineages across assay replicates. High fitness lineages (putative PPIs) are correlated, but not low fitness lineages (no PPI). **I)** The Pearson $R^2$ (y-axis) of the fitness of a PPIseq lineage across growth replicates, calculated for all lineages with fitnesses above a particular threshold (x-axis). Low fitness lineages (no PPI) appear uncorrelated, while high fitness lineages (putative PPIs) are well correlated. **J)** A Sankey diagram showing how the lineage fitness results were filtered to detect PPIs. We required a fitness call in at least three lineages representing the same ORF combination to call a PPI.

for these controls as well as all pairwise combinations amongst several human ORFs (TRIM15, AKT1, BCL2L11, Bad, Noxa, UBE2S, Mcl-1) and the ORF from yeast *LEU2* (Methods). As expected, none of the negative controls showed growth in the presence of MTX. However, we observed growth, indicating a likely PPI, for the Leu2p homodimer [66], the TRIM15 homodimer [67], and of BCL2L11 and Mcl-1 [68]. Five other potential interactions indicated in the Intact PPI database did not result in growth detectable in this plate-based assay (S15 Table), although the occurrence of false-negatives is consistent with other high-throughput PPI assays [17].

## Pooled generation of PPIseq libraries

The SARS-CoV-2 pandemic spurred the application of a variety of high-throughput PPI assays to map out vhPPIs of SARS-CoV-2 and human proteins. So to test and compare our approach to other genome-wide surveys we screened for PPIs between human and SARS-CoV-2 proteins. We used Gateway Entry plasmids as sources of human ORFs (hORFs) and viral ORFs

(vORFs) to the screen. We grew and pooled the human ORFeome v8.1 collection, and for SARS-CoV-2 ORFs we purchased the plasmid collection generated by Kim et al. [69]. This collection was generated from synthetic DNA and included 28 complete annotated coding sequences and 8 additional fragments or mutants thereof (S2 Table). hORFs and vORFs were moved from their pooled Gateway Entry vectors into our base PPIseq assay plasmids, the appropriate lox cis-element (loxP/66 or lox5171/66) and 30bp random DNA barcode was integrated, and this library was cloned at ~3-6x coverage of colonies per ORF. vORFs were inserted into the F[3]-tagged vector (pSL737, S5 File) and hORFs into the F[1,2]-tagged vector (pSL51 S4 File).

A crucial aspect of a barcode-sequencing approach is to efficiently map which barcode represents which genetic variant, so we sought to determine the barcode-ORF map by using long-read sequencing. We linearized plasmid pools using restriction digest (I-SceI) and generated sequencing datasets using both PacBio and Nanopore technologies. We devised a bioinformatic Nextflow pipeline to trim and extract the 30p barcode from these raw reads, separate out reads that contain each barcode, and use these reads to generate and polish a consensus ORF sequence integrated into the plasmid with this barcode. We generated a mapping of 9,191 F [1,2]-tagged hORFs represented by 35,191 barcodes and 36 F[3]-tagged vORFs represented by 608 barcodes, each with the ORF sequence verified as correct or a synonymous codon change (S4 and S5 Tables). By comparing the small Pacbio dataset and the more extensive Nanopore dataset for the barcoded hORF plasmids, we found that only four barcodes mapped to multiple ORFs amongst the >16,000 barcoded plasmids seen by both annotations runs, and excluded these four barcodes from downstream analysis.

We then integrated each plasmid pool into the appropriate yeast strain, choosing the MATa ySL507 for the vORFs tagged with F[3] and the MATalpha ySL508 for the hORFs tagged with F[1,2]. While our routine use of the published PEG/LiAc protocol [70] for yeast transformation was sufficient to integrate the small vORF library at high coverage (estimated as 25-196x yeast transformant colonies per barcoded vORF plasmid), we found that the efficiency of iSeq plasmid integration would be limiting for transforming the larger hORF plasmid library of >35,000 barcoded plasmids. We systematically tried variations on the published PEG/LiAc transformation protocol and found that recovering the heat-stressed cells in glucose-containing media without a nitrogen source was associated with a noticeable increase in transformants per ug of plasmid (S12 Fig), theoretically consistent with the role of nutrient shifts in triggering transporter endocytosis as part of the endocytosis hypothesis for the mechanism of PEG/LiAc transformation [71–75]. Using this modified protocol (S1 File) we found an approximately 2.5x fold improvement in transformant colonies per ug of plasmid, and were able to generate a library of approximately 337,000 genomic integrants of the barcoded F[1,2]-tagged hORF plasmids.

## Screening for human and SARS-CoV-2 ORF protein-protein interactions

With complimentary pools of yeast with tagged hORFs or tagged vORFs integrated into yeast genomic landing pads, we generated all-by-all combinations by pooled mating. We also generated pools of positive and negative controls by mating small (8–10 distinct barcoded isolates) haploid yeast pools with control constructs (S6 Table). Mating the pools of NULL-F[3] or NULL-F[1,2] haploid strains together generated double-negative controls (NULL-F[3] x NULL-F[1,2]) with no resistance to MTX, while a positive control was generated by integrating the F[1,2] tag into the F[3]-tagged expression context—generating a covalent F[1,2]-F[3] fusion with 100% co-localization of the mDHFR tags. The hORF and vORF pools were similarly mated to the NULL-F[1,2] or NULL-F[3] construct pool to detect promiscuous proteins

that interact with the complementary mDHFR tag directly. We induced recombination for each of these mated pools and selected for the double-barcode amplicon using the split *URA3* marker (Methods). We then combined the pools of diploid PPIseq assay strains for quantification of the selection. A challenge for barcode-sequencing screens is in balancing the library such that sufficient read depth is available to adequately quantify individual barcode abundance in highly complex libraries, even as those barcoded lineages change in abundance in response to selection-based assays [76]. Thus, we first cultured the negative control, positive control, and test strain pools separately for one cycle of MTX selection before combining the pools with a relatively small abundance of positive controls. We sampled this initial double-barcode lineage pool, then continued cycles of MTX and selection and sampling (Methods). The selection culturing and sampling of the lineage pool was done in triplicate.

From these samples, we used our modified DNA extraction and amplicon library generation protocols (S3 File) to generate double-barcode amplicon libraries for sequencing. We found quantification of the double-barcode lineages (Fig 2B) to be reliable, with two independent biological samples of the initial lineage pool being in good agreement ($R^2 = 0.976$ S4 Fig). In this first time point 211,688 possible combinations of ORFs were detected with at least three double-barcode replicate lineages each, representing at least 60.5% of the possible ORF combinations (Fig 1D, **for counts table see** Methods, **Data access**). We analyzed these abundance estimates with a fork of FitSeq [77, 78] to calculate per-lineage fitness estimates while accounting for the change in mean population fitness (Fig 2D, S7, S16 and S17 Tables). Comparing the fitness of each lineage (Fig 2F, 2G and 2H, scaled by the average fitnesses of the positive and negative control lineages Fig 2C) showed positive PPIs are reproducibly quantified ($R^2 > 0.9$ for lineages with >0.2 fitness, Fig 2I). In total we obtained ~1.5 million unique double-barcode lineages per replicate that we analyzed to call PPIs.

## Using double-barcoded lineage replicates to detect promiscuity, toxicity, and dropouts

The control lineages largely behaved as expected (Fig 2B and 2C), with the F[1,2]-F[3] (covalent tag fusion) construct showing the highest fitness and sweeping to high relative abundance, and the double-negative controls (NULL-F[1,2] x NULL-F[3]) showing low relative fitness. We tested for promiscuous proteins by analyzing the NULL-F[1,2] x vORF-F[3] or the hORF-F[1,2] x NULL-F[3] strains, and did not find any SARS-CoV-2 vORFs that mediate escape from the MTX selection by themselves. However, the F[1,2]-tagged human ORF of CSNK1G2 robustly escaped MTX selection when expressed along with a NULL-F[3] tag, and may reflect some interaction of human proteins with the yeast metabolic regulation of the native *DFR1* or a direct interaction of CSNK1G2 with the PPIseq assay constructs. We were also able to detect that expression of the F[1,2]-tagged human DHFR rescued growth from the mild (1ug/mL) MTX treatment we use here (Fig 2B and 2C), and this served as an additional positive control. We did not find any specific inhibition of yeast growth for particular vORFs (S7 Fig). However, we did not obtain any barcoded plasmid lineages for either wildtype or mutant NSP3 or the C-truncated version of ORF7B, but this may be explained by the fact that they are the longest and shortest ORFs in the collection (5835bp and 60bp, respectively). Examining these DHFR-F[1,2] positive controls, we unexpectedly found that a few lineages of positive controls would drop out, meaning that the lineage decreased in abundance and produced a low fitness (S5A Fig). Consistent dropouts of the same lineages across replicates (S5B Fig) indicated that the pattern wasn't simply due to sampling noise from culturing. Manual inspection ruled out a bioinformatics error as the explanation, so it became apparent that some lineages simply yielded a fitness discordant with the other lineages representing that

specific biological ORF combination. This was also true for some positive hit PPIs (S2 Fig). We suspect this problem is due to the frequent generation of petite-mutants in yeast, which will be addressed in the Discussion. To address these technical artifacts in our analysis pipeline, we took advantage of our platform's approach of generating many barcoded replicates for each genotype-of-interest (each ORF-ORF combination). We used a heuristic to drop lineages that were greater than 1 standard-deviation away from the mean for a particular ORF-ORF combination in at least two biological replicates, as long as this did not reduce the number of lineages to below three.

Another unexpected finding was that the positive-controls for the ORF expressing NSP2 (DHFR-F[1,2] x NSP2-F[3]) had a slightly higher fitness than other positive-controls, consistent across the replicate selections (S7 Fig), yet we did not detect any NSP2-specific fitness difference for the NULL-F[1,2] x NSP2-F[3] negative controls. This may reflect some subtle effect that NSP2 expression has on folate metabolism [79] in budding yeast or the mDHFR tags directly, so we mitigated the effect of this by subtracting the average NSP2-specific advantage from all NSP2-containing lineage fitnesses. While this effect appears to be limited to NSP2-containing strains, this adjustment may affect the False Discovery Rate correction for other ORF-ORF combinations.

## Detecting PPIs from a massively-multiplexed sequencing assay readout

To detect PPIs, we looked for sets of double-barcode lineages representing the same ORF-ORF combination with a higher than expected fitness. We unexpectedly found that NULL-F[1,2] x NULL-F[3] double-negative controls were on average more fit than either the hORF (hORF-F [1,2] x NULL-F[3]) or vORF (NULL-F[1,2] x vORF-F[3]) negative controls. We also found that the median fitness of the fitness distribution of all hORF-F[1,2] x vORF-F[3] test strains is similar to that of either hORF (hORF-F[1,2] x NULL-F[3]) or vORF (NULL-F[1,2] x vORF-F [3]) negative controls (Fig 2C). This appeared to be a general effect, as we did not find good evidence of any specific fitness cost associated with any individual hORF or vORF. Because of this effect, we chose to compare the fitnesses for each set of lineages (grouped by hORF and vORF) to the average fitness of the NULL-F[1,2] x vORF-F[3] negative control for that particular vORF.

We tested 184,491 groups of hORF-F[1,2] x vORF-F[3] lineages, requiring first that the group contains at least three lineages with fitness estimates (Fig 2J). We then used Welch's one-sided t-test with a False Discovery Rate (FDR, Benjamini Hochberg) correction to detect PPI hits (S18 Table). Across the three replicates at a FDR < 0.05 threshold we found 51 ORF combinations that produced hits in only one replicate, 13 that produced hits in only two replicates, and 10 combinations that produced hits in all three replicates. As shown in Fig 2J, ~53% (184,491) of the ~350k possible lineages (~330k when counting unique genes) are able to be assayed in at least one replicate. Thus, we detect putative PPIs for 0.04% of the testable ORF combinations.

## PPIseq complements previous human-SARS-CoV-2 PPI screens

We next compared our results to previously observed PPIs in the IMEx Coronavirus dataset [80] to determine the overlap with our 74 PPI hits or the 184,417 ORF combinations for which we did not detect a PPI (Fig 3A). We found a significant overlap (p-value < 0.05) of 8.1% of hits being previously observed compared with 0.75% of the combinations not called as a PPI. This confirms that the mDHFR split-tag assay reproduces detections by some other testing modalities (two-hybrid, GFP split-tag, or affinity purification), but also suggests that it has complementary sensitivities or biases to help complete a better understanding of PPI networks

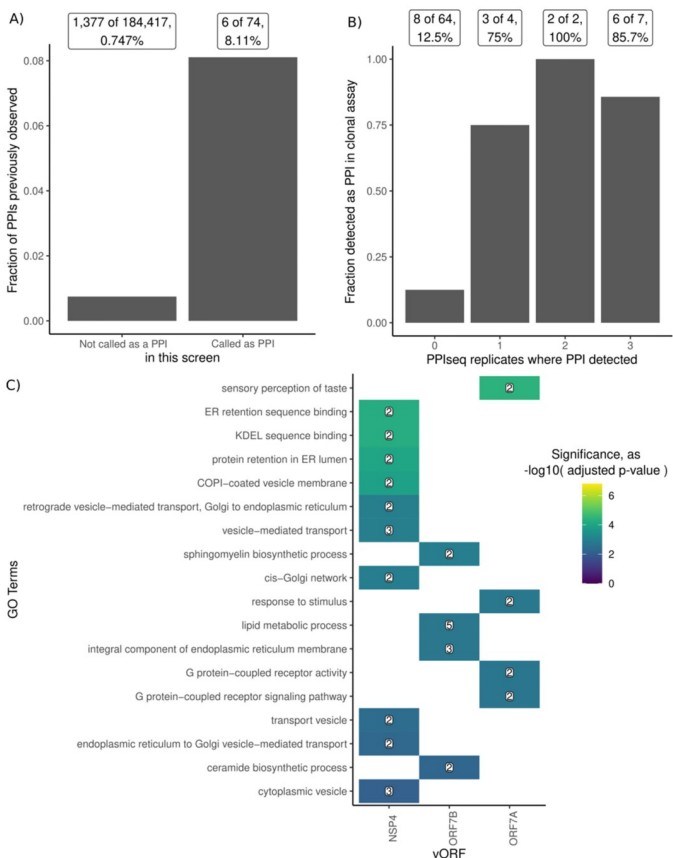

**Fig 3. PPIseq confirms previously detected human-SARS-CoV-2 PPIs and finds new interactions involving taste receptors and ER-retention. A)** The fraction of protein pairs screened by PPIseq that have been previously reported as PPIs in the IntAct Coronavirus dataset. Protein pairs are partitioned into those that have not (left) and have (right) been called as PPIs by PPIseq. **B)** The fraction of protein pairs that were validated as a PPI in a clonal growth assay when 0, 1, 2, or 3 of the PPiSeq growth replicates detected a PPI. **C)** GO term enrichment analysis (using clusterProfiler [82]) of human proteins that interact with the products of viral genes NSP4, ORF7B, and ORF7A. The color of each tile is the negative log10 of the adjusted p-value for the GO term enrichment, all shown are adjusted p-value < 0.05. The number in each tile is the number of human ORFs with that annotation in the set.

[17]. To validate the replicability of PPIseq in quantifying the mDHFR split-tag assay we repeated the mDHFR assay using a low-throughput clonal measurement of colony growth on agar. We selected human and viral ORFs that were constituents of several detected PPIs and regenerated 11 clones of haploid yeast with each human ORF integrated in a mDHFR-tagged expression locus. To obtain the 7 complementary vORF-F[3] haploid yeast strains we used REDIseq [81] (Methods), then we generated each combination of hORF-F[1,2] x vORF-F[3] diploid strains by pairwise mating and selection as before. Scans of the resulting MTX growth assay of the diploids (S19 Table) were analyzed to quantify colony size. Despite assaying the strains on agar media and with 100-fold larger concentrations of MTX (necessary to generate an effect large enough for visual analysis), we found 11 of 13 PPIs amongst these ORFs were detected again (Fig 3B). By contrast, 8 of 64 background combinations produced detectable growth and are examples of false negatives in the PPIseq assay—likely due to insufficient coverage of lineages per PPI in the selection pool (Discussion). Although false negatives are pervasive in PPI measurement assays, this shows that our PPIseq quantification of the mDHFR split-tag assay is consistent with the performance of other platforms, detects a fraction of the

PPIs that have been previously observed, and also detects PPIs that may be inaccessible by other techniques. These new detections are likely by virtue of the mDHFR assay's compatibility with membrane-bound proteins, diverse organelle localization, and proteins that are poorly expressed in human cell culture lines or poorly purified.

To generate hypotheses of how these interactions might contribute to the functional roles of each viral ORF, we analyzed the properties of the human ORFs detected as PPI hits with each viral ORF. We calculated enrichment using a hypergeometric test or the hypergeometric-distribution-based enrichment test, as implemented in clusterProfiler [82], for GO terms, having transmembrane domains, PROSITE domain annotations, and Reactome pathway annotations. Here we find that the human proteins of PPI hits for two viral proteins (NSP4, ORF7B), are significantly enriched (p-value < 0.05) for interactions with proteins containing transmembrane domains (by Uniprot annotation, 5/6, and 11/12 respectively), while ORF6 and ORF7A were only detected with transmembrane proteins. This is consistent with ORF7B partners being enriched for GO terms related to lipid metabolism and endoplasmic reticulum membrane (Fig 3C), although this may also reflect a hypothesized promiscuity of the proposed leucine-zipper motif in binding other transmembrane proteins [83]. ORF7A was only found to interact with the taste receptors TAS2R41 and TAS2R7, which is intriguing considering the genetic evidence of variant-of-concern-specific ORF7AB alleles in a hamster-model of anosmia [84] and TAS2R41 being identified as a potential host-factor in a loss-of-function screen in cell-culture [85]. We confirmed the previous detection of ORF6 interacting with ABHD16A, a mechanism that may supplement ORF6's antagonism of cellular innate immunity [86–89]. NSP4 was found here to interact with KDEL-receptors (both KDELR2 and KDELR3), and this interaction with the ER-retention retrograde signaling pathway might shed light on the role this protein plays in shaping ER membrane into the double membrane vesicles important for viral replication [90–92]. NSP2 was enriched (p-value < 0.05) for interactions with proteins annotated with the Reactome pathway of Smooth Muscle Contraction, including the Calmodulin CALM3 and Calmodulin-binding protein CALD1. These analyses show that the complementary sensitivities of the mDHFR, specifically to PPIs of transmembrane proteins involved in the cellular membrane biology critical for some viral processes, can help us identify more complete hypotheses of how viral proteins interact with host factors.

## Discussion

We extend the capabilities of the PPIseq platform to efficiently screen for protein-protein interactions (PPIs) using pooled methods for plasmid engineering, DNA barcoding, annotation, library generation, and selection. In this work, one researcher was able to use the PPIseq platform to screen at least 55.8% of the possible PPIs between the SAR-CoV-2 and human proteins in a library of >330,000 protein pairs, finding 74 hits. Six of these hits were previously observed in the IMEx Coronavirus dataset and 68 are new. PPIseq uses the mDHFR split-tag as our PPI selection system, which has high sensitivity to transmembrane protein PPIs and serves to complement previous SARS-CoV-2-human PPI screens. In particular, we found interactions that suggest a potential role for ORF7A in mediating interactions with human taste receptors and NSP4 in mediating ER retention via the KDEL-receptors, interactions that might be critical for function during infection but are likely challenging to detect using other split-tag reporters or mass-spectrometry-based approaches.

A key advantage to PPIseq is scale. We used a pooled approach whenever possible, which lowers the cost and hands-on time per PPI assayed relative to array-based alternatives. Another advantage is our use of whole-plasmid sequencing of barcode-ORF pairs, which detects any errors in the ORF sequence. ORF sequence errors can be a problem for libraries of arrayed

constructs that can accumulate mutations or be cross-contaminated between positions. Another advantage of the all-pooled approach is that the methods here do not require specialized equipment. High-throughput sequencing is available commercially, and the provided protocols (Supplement) and code (Methods) can be used with typical microbiology lab equipment to screen other Gateway ORF collections.

However, one major caveat of our pooled approach is that we are unable to balance the abundance of each species of the pool, resulting in a skewed abundance distribution (S11 Fig). Future use of this workflow may benefit from isolating arrays of clones to rebalance the relative abundance of haploid strains before mating them to generate the diploid double-barcode PPI-seq assay library. While isolating arrays of clones is generally a laborious undertaking, the elaboration on the PPIseq platform described here yields a PPIseq haploid library that is compatible with the iSeq 2.0 platform [51] and thus can use REDI-seq to rapidly identify the identities of randomly arrayed clones [81]. We tested this indexing capability here by randomly arraying the pool of barcoded haploid tagged-vORF yeast into an arrayed library and using REDI-seq to annotate the identity of each position (S10 Fig). The strains from this array were used for the PPI verification shown in Fig 3B. This approach, or alternative technologies for "indexing" or "demultiplexing" a mixed pool of plasmids [93], offer opportunities for efficiently separating and balancing the abundance of constructs before combining again in pools. Such an approach may be valuable when reducing the all-by-all test pool to smaller subsets of test strains that are of interest, such as when screening a set of PPIs for small-molecule disruption.

There are also limitations to our split-tag reporter assay, and split-tag reporter assays in general, that could contribute to a high false-positive or false-negative rate. We have not verified that each of the thousands of ORFs integrated into the tag-fusion construct are faithfully and accurately expressed in each strain, that the expression of each tagged-ORF is similar, or that they fold to generate all functional domains relevant for an interaction. Each of these proteins may also be missing post-translational modifications from host or pathogen factors, or subject to aberrant post-translational modification when expressed in budding yeast. Obligatory tertiary binding partners may be absent, or promiscuous binding could be mediated by a partner present in yeast but not the natural host cells [94]. Additionally, the mDHFR split-tag may interfere with a functional binding interface or prevent proper co-localization. The seemingly low overlap with previously reported PPIs (6 of 74) is consistent with the difficulty of reproducible detection while limiting false-positives in the massive search space of PPI screens [89], but this also likely reflects that the mDHFR split-tag suffers from the same kinds of limitation and bias as other assay modalities. While the mDHFR split-tag reporter may be more sensitive to membrane-associated proteins it is also likely biased against detecting PPIs of nuclear proteins [17], for example. The expression of ORFs may also modify yeast physiology to prevent a faithful increase in batch culture fitness as a result of the mDHFR assay. For example, 5.6% of approximately 10,000 human cDNAs were found to be toxic when over-expressed in yeast [95]. Despite these limitations, the significant overlap with previous SARS-CoV-2-human screens and the established performance of the mDHFR-tag suggests that PPIseq is a viable platform for scaling the mDHFR split-tag assay.

Besides the above limitations that are common to most split-reporter screens, other technical limitations of the method as realized here are more readily addressable. For example, we observed lineage-specific "dropouts" where a few lineages that represent a particular protein pair would decrease in relative abundance (despite an average high fitness), but never the converse where rare lineages would have an increased fitness. The consistency of lineage fitness for dropouts across replicates suggests this is not a function of noisy genetic drift due to limited cell bottlenecks, and instead may be caused by a mutation that reduces growth rate. A likely

candidate is "petite" mutations, a common class of mutation in lab strains of budding yeast that results in the loss of the mitochondrial function and slow growth. If this is the cause, future work could pre-culture strains on a non-fermentable carbon source or use a "repaired" lab strain with a lower rate of petite formation [81]. There are also opportunities to improve statistical analysis. Here we used a statistical approach similar to our previous work [51] wherein we test if the sample of fitnesses for the group of lineages representing a particular ORF-ORF combination are greater than a null distribution. More sophisticated statistical methods that can confidently detect per-lineage or per-group fitness changes while moderating the variance ubiquitous to the use of sequencing platforms for quantification. For example, the FitSeq-derived population average fitness can be used to adjust library scaling parameters in a tool such as the limma/voom differential expression analysis pipeline [96], and rigorous development and testing of these approaches could enable detecting changes in single lineage fitnesses between treatments. Moreover, non-linear modeling of the signal could be pursued to enable more biophysically interpretable results [97].

Decades of technology development has enabled foundational genome-scale studies of PPI networks and how they evolve [28, 32, 46, 47, 80, 89, 98, 99]. This paper builds on that work by enabling an all-pooled library-generation and measurement workflow for PPIseq, contributing to the diversity of genome-scale PPI assays [100] with a workflow accessible to non-specialist research groups. Applying these methods to characterize virus-host PPI networks for many virus species and strains will help better describe the dynamics of PPI network evolution, and the use of pooled tagged-ORF plasmid libraries as inputs can leverage high-throughput pooled reverse genetic tools to dissect the genetic contributions to altered PPIs [101]. PPIseq is based on an interaction-barcoding platform that can also be readily adapted to screen alternative split-tags [102] and tag-orientations, or to assay toxicity or extracellular interactions [103, 104], and together these technologies can help test foundational concepts of the mutational supply of PPI alterations, their effects on fitness, and the resulting dynamics of PPI network evolution. This path of combining PPI network surveys with measurements of the cause and consequence of altered PPIs will permit an evolutionary systems biology analysis of how the accessibility or "roughness" of mutational paths across a fitness landscape and the ecology of viral/host systems interact to manifest biological complexity in the structure of PPI networks [105].

## Materials and methods

### Code access

Computer code used to analyze data and generate graphics is available in several git repositories. For the analysis of the barcode annotation datasets, see https://gitlab.com/darachm/ppiseq_dme353/ for analysis of the first PacBio dataset as part of Fig 1, then see https://gitlab.com/darachm/ppiseq_d368/ for analysis that extends this mapping using Nanopore dataset to generate the barcode annotation map finally used for Fig 1. For analysis of the PPIseq selection data and/or REDIseq array identification data (three lanes of Illumina HiSeq) for Fig 2, see https://gitlab.com/darachm/ppiseq_d359d360. For analysis of the PPIseq hits used to generate the work surrounding Fig 3, see the analysis at https://gitlab.com/darachm/ppiseq_d359bio. For analysis of the plate images for the re-assay on agar media (Fig 3), see https://gitlab.com/darachm/ppiseq_dme383/. For each, additional associated "data" files are available at OSF (doi.org/10.17605/OSF.IO/B8G3H), as well as ZIP-ed archives of each git repository in case the hosted repositories are unavailable. Direct any technical questions to Darach Miller using the 'Issues' feature on the GitLab website.

## Data access

Sequencing datasets are available on the Sequence Read Archive. For long-read sequencing (PacBio and Nanopore) used to annotate DNA barcodes with the linked ORF they represent, see PRJNA1073210. For Illumina sequencing of the double-barcode amplicon used to quantify the fitness of each diploid lineage, see PRJNA1073201. See the supplemental table (S8 Table) for the more metadata to contextualize these. For other intermediate files see the supplemental files for this paper, or retrieve them as a sqlite3 database file from this OSF repository (doi.org/10.17605/OSF.IO/B8G3H). For example, because of the particularly large size of the counts table, that table is available as a separate sqlite3 database in the linked OSF repository. Raw TIFF scans of the agar plates used for validation of the modified mDHFR assay (Fig 1) or PPI re-testing on agar media (analyzed for Fig 3), is available in the above OSF repository in an appropriate folder.

## Yeast media and strain construction

Yeast media was made as described in the 2015 edition of the CSHL Yeast Course manual [106], specifically YPD, SC (synthetic with complete amino-acid supplementation), and SD (synthetic dropout with only specific amino-acids supplemented back). Dextrose, yeast-extract, peptone, and yeast nitrogen base (without amino-acids or ammonium sulfate) were supplied by BD, and supplemented by ammonium sulfate from Sigma. Amino-acid additive stocks and selective plates with 5-FOA (GoldBio), hygromycin, or G418 were made as in CSHL Yeast Course manual.

Previously engineered BY4741/4742-background *Saccharomyces cerevisiae* strains contain-ing iSeq Gal-Cre landing pads ySL167/XLY5 (MATa) and ySL173/XLY11 (MATalpha) [52] were engineered further to delete the coding sequence of *PDR1*, *PDR3*, *PDR5*, and *SNQ2*. We used the 50:50 method [107] several times to do this. Briefly, we designed primers that amplify a *URA3*MX3 cassette from pRS426 [108] with flanking sequence-identity that mediates homol-ogous-recombination into the yeast genome. This insertion cassette is selected on SC-URA for the presence of the *URA3*MX3 marker, but the cassette is designed such that it also introduces an identical sequence repeat that spans both the *URA3*MX3 cassette and the target coding sequence. We expand these URA+ colonies in YPD liquid media overnight, then plate this population on SC+5-FOA to select for clones where the identical repeat has promoted excision of both the marker and the target coding sequence. Colonies are restruck on SC+5-FOA and clones from this are screened by colony PCR. Deletions were carried out in series, then qualita-tively checked for function (growth on non-fermentable carbon source, integration of iSeq plasmids, mating, and function of MTX selection using positive/negative control constructs as described in the Controls Strains section). A clone of each type were saved as strains ySL507 (MATa, loxP/71) and ySL508 (MATalpha, lox5171/71).

## Construction of PPiSeq base plasmids

Previously published plasmids pBAR4 and pBAR5 [52] were further engineered (using typical methods of digestion, ligation, Gibson assembly, and cloning in *E. coli*) to integrate a Gateway cassette (attR1-ccdB-CmR-attR2) for expressing a Gateway-integrated ORF as a fusion to either a F[1,2] or F[3] mDHFR split-reporter tag [28]. Gateway recombination from an attL1-attL2 Gateway Entry vector results in the ORF being fused in frame with the standard Gateway scar, then a 4xGGGGS linker [53], and then either of the two mDHFR reporter tags. The promoter of this expression construct is the common TDH3pr / pGPD promoter, and the terminator is a 35bp minimal terminator (Synthetic 7) as designed and tested by Curran et al 2015 [59]. Immediately downstream of the terminator is an I-SceI site. The rest of the plasmid

is derived from pBAR4/pBAR5, and so each of the plasmids has a complementary intron donor or branch site [52, 109], adjacent to that either *ura3*ΔC or *ura3*ΔN with a promoter and terminator as appropriate, either a KanMX or HygMX drug-selectable yeast marker, and a ColE1 origin and AmpR marker for propagation in *E. coli*. Importantly, these base plasmids do not contain any lox sites, barcodes, or amplicon-sequencing priming sites. These plasmids were saved as plasmid-bearing clonal strains of *E. coli*, with plasmids pSL51 bearing the F[1,2] mDHFR tag and pSL737 bearing the F[3] tag.

## Entry vector (Gateway) plasmid pool sources

The human ORFeome (hORF) v8.1 collection (purchased from Dharmacon, https://horizon discovery.com/en/gene-modulation/overexpression/cdna-and-orfs/products/horfeome-v8-1 -collection) was pinned from stocks onto LB+Spectinomycin agar, and colonies were collected by scraping into 9 sub-pools of up to 16 96-position plates. These pellets were resuspended and frozen in 15% glycerol, and a frozen aliquot of each was later collected and plasmids were extracted via miniprep kit (Qiagen).

Vectors with ORFs that encode proteins similar to those encoded by SARS-Cov-2 (vORFs) were obtained as single clonal E. coli plasmid-propagating strains from Addgene, and are a generous gift from the Roth laboratory by way of Addgene [69]. The identity of each clone was verified with partial Sanger sequencing. ORF-containing plasmids were pooled into 3 sub-pools, with membership in each pool being arranged such that no pool contains more than half of the genome (by total length). These ORFs were handled in separate pools at all stages until the MTX-selection competition assay of diploid yeast libraries, or after preparing bar-code-indexed libraries for PacBio or ONT Nanopore sequencing.

## ORF integration and tagged ORF plasmid linearization

Each subpool (9 hORF or 3 vORF subpools) of plasmids was combined with either pSL51 (hORF) or pSL737 (vORF), TE (Tris-EDTA buffer), 5x reaction buffer, and LR recombination enzymes from the Gateway LR Clonase kit (Thermo) as according to manufacturer instructions. Reactions for hORFs were 20uL total volume with ~300ng each plasmid pool / destination vector, and 10uL total volume with ~150ng each plasmid pool / destination vector for vORFs. The vORF reaction was left at RT for 1 hour, digested with proteinase K, 2uL was transformed into 10beta Mix-and-Go chemicompetent cells (Zymo), and cells were plated to select on LB+Carb plates. The hORF reaction was left at 25°C for 24 hours, treated with proteinase K, concentrated with an ethanol precipitation, electroporated into 10beta electrocompetent cells (NEB), then the 1-hour-recovered cells were plated to select on LB+Carb plates. Cell counts were estimated from a plated dilution and the rest of the colonies were scraped to collect and freeze cells at -80°C. An aliquot of each subpool was miniprepped to extract out the LR-recombined tagged-ORF plasmid pools.

## Integration of the lox site, random barcode, and priming site cassette

Each "loxcode" cassette is a 130-138bp dsDNA fragment that contains: sequence-identity to the terminator, loxP/66 or lox5171/66 sites (cis-elements for Cre/lox recombination), a 30bp random barcode (IDT, N_30 machine mix), priming sites for amplicon sequencing, and an I-SceI site. For the lox5171 loxcode cassette we re-used the iSeq reverse priming site [52] with some modifications (here calling the priming site "iseq2R"), but for the loxP fragment we devised a novel priming site based on a synthetic spike-in [110] that was manually adjusted based on personal experience (here calling the priming site "mjtF").

The dsDNA loxcode cassettes were formed by combining two purchased oligos (IDT) in a 50uL Q5 (NEB) polymerase reaction with the Q5 polymerase, 200mM dNTPs, 500nM primers, and 1x Q5 buffer. Each reaction is run on a thermocycler to melt at 98˚C for 30s, brought to 70˚C for 1s, then cooled to 40˚C at 0.1˚C/s rate, then brought to 72˚C for 20s. Each reaction is then cleaned using a DNA Clean and Concentration kit (Zymo) at a 5:1 buffer:sample ratio, and eluted with 20uL EB and frozen. oSL1186 was combined with oSL1187 to generate the loxP/66-barcode30-mjtF-ISceI loxcode cassette, and oSL1159 was combined with oSL1158 to generate the lox5161/66-barcode30-iseq2R-ISceI loxcode cassette.

Each tagged-ORF plasmid subpool was treated with I-SceI (NEB) in the provided buffer (at a volume of 50uL for ~1ug hORF plasmids and 200uL for ~4ug vORF plasmids). This reaction was incubated at 37˚C at least overnight, then was treated at 65˚C for 30min to denature the I-SceI (a critical step). This was cooled and treated with phosphatase rSAP (NEB) at 37˚C for 30min, then denatured again at 65˚C for 10min before ethanol purification. With this approach test samples appeared to yield complete linearization as viewed on an agarose gel and this procedure was important to minimize self-closure in the next cloning step.

Approximately 6ng of each loxcode cassette was combined with approximately 75ng of the linear digested plasmid pool in a final 10uL of 1x HiFi assembly reaction (NEB). This was incubated on a thermocycler at 50˚C for 15min, then put on ice. The reaction was ethanol precipitated, then resuspended in water. A dilution of each reaction, as determined based on empirical tests of colony yields, was electroporated into 10beta electrocompetent cells (NEB). After recovery, these were plated, grown overnight, and approximately 2-4x coverage of the ORFs expected to be in the pool were scraped into a cell pool, yielding each E. coli pool containing barcoded tagged-ORF PPiSeq assay plasmids. Aliquots were frozen in 15% glycerol, and one aliquot of each pool was extracted using a Qiagen miniprep kit.

After the barcode-ORF map was determined (see below), we thawed frozen aliquots of selected pools of E. coli containing barcoded-ORF plasmids, the same pools from which the barcode-ORF map data was generated, and diluted these in LB+Carb liquid media. For the three vORF pools these back dilutions were grown to approximately exponential phase growth and Qiagen miniprepped. For the hORF pools, these were each diluted 1/20 and incubated with shaking for approximately 3 hours at 37˚C and confirmed to reach about saturation (by OD). These starters were then back diluted again approximately 1 in 200 into LB+Carb liquid media. After 2 hours, OD for each pool was used to combine the hORF pools with volume determined by the library complexity (ie as the number of unique plasmids found in the Pac-Bio barcode-ORF sequencing) and this pool were cultured at 37˚C overnight (15 hours) after the addition of chloramphenicol to a concentration of 170ug/mL. In the morning, the culture was collected by centrifugation and plasmid pool was extracted with a Qiagen maxiprep kit in two batches. During this entire process, each vORF pool was kept separate, but all 9 of the hORF subpools were combined before maxi prep.

## Testing the plasmid-based mDHFR-tag PPiSeq system and constructing control strains

To facilitate the construction process we first took the base expression plasmids (pSL51 and pSL737) and integrated the lox cis-element, random barcode, and priming site cassette as described above—importantly these plasmids still contained the Gateway ccdB counter-selection landing site but the pool of generated plasmids also contained a complex barcode library at the barcode locus. This is the opposite order as used in the large-scale experiment (where ORF is integrated first, then barcode), but is easier for engineering of single clones of plasmids. We isolated ORF-bearing Entry plasmids from clones in the arrayed collection of the Gateway

entry vectors for the human ORFeome v8.1 collection (purchased from Dharmacon). We used a Gateway LR Clonase mix (Invitrogen) to move the ORFs from these plasmids into the pre-barcoded pSL51 and pSL737 plasmid pools, and cloned these into 10beta cells made competent via the Mix-and-go kit (Zymo Research). Clones from these were arrayed in a 96-well plate and screened for chloramphenicol resistance that would indicate presence of the Gateway counterselection cassette, then screened using colony PCR of oligos oSL231 against oSL261 on the pSL737-based plasmid and oSL231 against oSL260 on pSL51-based plasmid. These were then Sanger sequenced to confirm the identity of the ORF integrated—for example we found that two attempts at isolating plasmid from position GDEh81037@B04 identified the ORF UBE2S instead of the ORF BCL2L1. These plasmids were all transformed using a lithium-acetate transformation protocol, as described in CSHL Yeast Manual [106], into the yeast strains ySL167 and ySL173, as appropriate given the loxP or lox5171 landing pad present in the genome, and selected using the drug markers on the integrated plasmids. Clones of these strains were picked into a 96-well plate and crossed by mixing in YPD overnight. Diploids were selected with YPD+G418+Hyg, then these were pinned onto YPGal agar plates for overnight induction before recombinant diploids were selected on SC-URA agar plates. These were pinned and grown again in 96-well SC-URA liquid media, then pinned to SC-URA +MTX (100ug/mL) agar media. After two days of growth this plate was imaged with a flat-plate document scanner.

We generated two types of control plasmids for this tagged-ORF expression system, a covalently-fused co-localization positive control and a NULL negative control of passive co-localization. For the first type of control, we generated a Gateway entry vector with attL1/2 sites flanking a F[1,2] mDHFR tag fused on its C-terminus to a 2xGGGGS linker, and this was integrated into the pre-barcoded assay plasmids (barcoded pSL51 and barcoded pSL737). Integration into the pre-barcoded pSL51 yields a mDHFR F[1,2]-linker-F[1,2] fusion that does not grow readily upon MTX treatment (a negative control), while integration into pre-barcoded pSL737 yields a mDHFR F[1,2]-linker-F[3] fusion that rescues growth upon MTX treatment (positive control). For the second type of NULL negative control, we directly engineered the pre-barcoded pSL51 and pre-barcoded pSL737 plasmid pools using PCR and HiFi *in vitro* recombinational assembly (NEB reagents) to generate a construct that should express a single methionine start-codon fused to the Gateway attL2/attR2 "scar" sequence (YPAFLYKVV), the 4xGGGGS linker, and each of the two mDHFR tags. Even when combined, these "NULL" mDHFR tags untethered to any ORF did not rescue growth on MTX that we could detect given our experimental design, thus serving as a control for the subtle confounding effects of passive co-localization [65].

## Mapping the ORF-barcode annotation

To generate the barcode-to-ORF annotation mapping vital for using a barcode-sequencing approach, We digested ~2ug of each of 24 barcoded ORF plasmid pools (2 replicate pools of each of 9 hORFs and 3 vORFs) overnight with I-SceI (NEB) according to manufacturer's recommendations, then denatured the reactions at 65°C for 30min before purifying each individual reaction using a phenol-chloroform reaction and ethanol precipitation. Each digested plasmid pool was subject to PacBio library prep using the SMRTbell Express Template Prep Kit 2.0 and SMRTbell Enzyme Clean Up Kit, as per manufacturer instructions (101-646-100 v7 and 101-800-100). Plasmid pools were inspected with a Bioanalyzer DNA 12,000 chip to confirm the absence of primers and approximate library size. These libraries were pooled as a multiplexed library and submitted the University of Arizona core facility for PacBio sequencing on the Sequel II platform.

This PacBio run yielded a high fraction of truncated reads, so we tried another platform. We pooled a set of 12 digested plasmids that had been miniprepped as part of their isolation, then prepared this pool using the ONT Nanopore Minion system with the ONT Native Barcoding (24) and Ligation Sequencing kits as per manufacturer instructions for a small-fragment sample and sequenced it on a R9.4.1 Minion flow cell. We observed the qualitatively the same pattern of read truncation as had been seen on the PacBio platform, so we revised our methods to include an Ampure XP bead cleanup step (0.6x) after the I-SceI digestion, then prepared libraries for ONT using the FFPE treatment and UltraII End Repair (NEB) as described in Nanopore library preparation protocols for genomic DNA. These performed well on a R9.4.1 Flongle, so we also sequenced these libraries using an entire R10.4 Minion flowcell.

To analyze these datasets, we devised a bioinformatics pipeline that (1) extracts the barcodes from the plasmid read, (2) trims away the backbone, (3) does a Multiple Sequence Alignment -based assembly of a draft consensus, and (4) uses Racon and Medaka to polish the assembly. This was done for both the PacBio (fewer reads, high quality) and Nanopore data (many reads, low quality), and the map of barcode to ORF, as well as any mutations in the ORFs detected, was compared (see Results). These maps were unified to determine what ORF and mutational-status is represented by a barcode (S4 and S5 Tables).

## Transformation and integration of barcoded tagged-ORF plasmids into landing pads

To rapidly express the Cre recombinase without a non-glucose preculture that would inhibit the transformability of the yeast, we used a transient transformation approach. Briefly, we amplified a TEF promoter and Cre recombinase coding sequence cassette using primers oSL331 and oSL702 from a template of pSL57 (55°C annealing and 2min extension), using OneTaq polymerase (NEB). This was purified on a McGeachy PCR Purification column.

For the barcoded vORF plasmid library the assay yeast strain ySL507 (MATa, loxP) was grown overnight in YPD, then back diluted and grown to ~0.4 OD in YPD. 300mL of this was collected in 50mL conicals with spins at 1000g for 3min, and 1mL of ice cold water was used to pool these into three tubes. These were collected by centrifugation (at 800g 1min) and washed with 500uL cold 0.1M LiAc, before resuspending with 20uL cold water and pooling into one tube. By hemacytometer this was 1.84e8 cells per 20uL. 20uL of these cells were put onto 180uL of a transformation mix (6uL 10mg/mL ssDNA, 140uL 50% PEG 3350, 21uL 1M LiAc, 2.1uL 1M DTT, ~500ng of TEF-Cre expression cassette, 5ug of each of three vORF plasmid pools, and water to bring to 180uL total volume). These were vortexed to mix, put at 30°C for 30min, 42°C for 30min, and collected by centrifugation. This was resuspended with 1mL of YPGal to recover, then spun and resuspended in 4mL YPGal after 1 hour and put at 30°C on a roller drum overnight. By hemacytometer, the induction cultures about doubled in the number of cells. The induced cells were then plated, and after growing colonies were counted and scraped into water suspension. By colony count, these had from between 25-196x coverage of colonies per unique plasmid in that subpool. This cell pool was brought to 15% glycerol and frozen at -80°C for later use.

The much more complex barcoded hORF plasmid library posed a challenge, as the yield achievable by the above protocol would require a laborious and expensive amount of plasmid and repetition. So, we devised adaptations of the lithium acetate and PEG protocol of Gietz et al. [70] to increase transformation efficiency, and used this to transform and integrate the barcoded tagged-ORF plasmids. An important modification is the gentle recovery of yeast, after the 65°C heat-stress, using YNB+Dex—media that contains "Yeast Nitrogen Base" and 2% glucose, critically without any nitrogen source. Using this protocol (S1 File) we were able

to generate a library of ~337,000 genomic integrants of the barcoded F[1,2]-tagged hORF plasmids. To do this, the assay yeast strain ySL508 (MATalpha, lox5171/77) was grown overnight in 2x YPD. In the morning, this was diluted with 2x YPD and grown for approximately 5 hours until reaching 0.8–0.9 OD. 400mL of this yeast culture was collected in 50mL conicals (3000g 5min) by centrifugation. These were washed twice, resuspending in 25mL and 10mL water and centrifuging, before another centrifugation and removal of all possible water from the pellets. During steps the yeast in the eight conicals were pooled to two conicals. These pellets were loosened by vigorous tapping and vortexing, before adding 10 volumes of the TRAFO mix (1 volume is: 240uL PEG 3350, 36uL 1M LiAc, 50uL of freshly denatured 10mg/mL salmon sperm DNA, 2ug of barcoded tagged-hORF plasmid pool, 500ng of the TEF-Cre transient expression cassette, and water to bring to a total of 360uL). Cell pellets were vigorously resuspended in this with a sereological pipette and vortexing, then put in a 42˚C waterbath for 70 minutes with agitation of the waterbath in the first 5min and inversion of the conical tubes to mix at 15min, 35min, and 55min. Pelleted cells at 3000g 5min, room temperature, then aspirated supernatant completely, using a second brief spin to remove all. The pellet was loosened again and then resuspended in 10mL (per conical tube) of YNB+Dex ("Yeast Nitrogen Base" without ammonium sulfate or any amino acids, supplemented with 2% glucose) and incubated at 30˚C shaking for 2 hours. This was pelleted again at 3000g for 5min, with supernatant aspirated, washed once with 5mL of YNB+Gal ("Yeast Nitrogen Base" without ammonium sulfate or any amino acids, supplemented with 2% galactose) before resuspension in 10mL of YNB+Gal and incubated overnight (approximately 16 hours) at 30˚C. During this step, no statistically significant increase in cell numbers was detected by hemacytometer counts. The cells were collected by centrifugation and resuspended in YNB+Dex before further dilution and plating on YPD+G418 (400mg/L) agar plates. This library spanned 39 plates, in addition to the dilution plates used for counting. By this method, the library was estimated to have approximately 337,000 integrations of plasmids into the ySL508 background. All the 39 plates were scraped and collected into a pool at 15% glycerol with approximately ~1.8e9 cells per mL, and frozen at -80˚C.

## Mating haploid yeast to generate a diploid assay library

We combined haploid yeast pools to mate and form double-barcoded diploid yeast lineages expressing two tagged-ORF constructs each. To do this, we first thawed and cultured each haploid yeast pool in YPD with the appropriate drug, using 100ug/mL hygromycin for yLSL47 (mDHFR F[1,2] positive control), yLSL48 (NULL negative control), yLSL52, yLSL53, and yLSL54 (the three SARS-CoV-2-like ORF pools) and 400ug/mL G418 for yLSL49 (NULL negative control) and yLSL55 (human ORF pool). We collected these overnight 30˚C cultures by centrifugation, then washed the cells with YPD and resuspended them at approximately 2e8 cells per mL density. We then mixed pools for each of the matings. Briefly we aimed to combine 1,000 times more cells of each pool than we expected there to be unique combinations in the mating library. These were combined and brought up to approximately 4e7 cells per mL density (2e7 cells per each pool per mL) in YPD at 10% glucose content. These were shaken at 30˚C for 4 hours, or incubated on the roller drum for small volume low-complexity libraries. Cells were collected by centrifugation and washed once with 1x YP (no glucose) media, then resuspended in nitrogen-source-less 1x YNB ("Yeast Nitrogen Base" without the nitrogen) supplemented with 2% Galactose. These were incubated overnight at 30˚C with shaking or roller-drum agitation, then plated on SC-URA plates at approximately 1e9 cells per plate density. After allowing the lawns to grow 2 days, these were scraped and collected into liquid SD+His +Leu media and expanded to culture approximately 200 cells per each lineage expected in the

pool. These were grown two days at 30˚C shaking, then diluted 1/10 again onto SD+his+leu media and grown two more days to select for URA+ double-barcode diploids.

## Culturing for the methotrexate (MTX) selection of diploid strain pools

To select on the MTX-treated growth phenotype of the pool of lineages, we diluted the diploid cell pools (still separated into three pools of vORF-hORF test strains, one pool of hORF-NULL negative strains, three separate NULL-vORF negative strains, and separate pools of NULL-NULL and positive controls) as 1 in 8 volumes back dilution into SD+His+Leu+1ug/mL MTX (methotrexate). These were grown two days at 30˚C shaking. The pools were then combined into one test pool and samples were taken for the first timepoint. The singular pool of strains was split into three, back-diluting 1 in 8 into SD+His+Leu+1ug/mL MTX and grown at 30˚C shaking in three separate flasks. This back-dilution, sampling, and culturing procedure was repeated several more times for each of the three separate culture replicates, ultimately collecting more samples than were processed for sequencing. Samples were taken by centrifuging 50mL of saturated liquid culture (grown in SD+His+Leu+1ug/mL MTX), completely removing the supernatant, dislodging the pellet by tapping, and freezing at -20˚C.

## DNA extraction for amplicon sequencing

We devised a protocol (S2 File) for DNA extraction that permitted large inputs of cells without excessive inhibition of the downstream PCR, and used this to process the samples for double-barcode amplicon sequencing. We prepared Xbuffer by dissolving into 30mL water (at 65˚C) 1g polyvinylpyrrolidone, 1g CTAB, 4.09g NaCl, 5mL of 1M Tris-HCl (pH 7.5), and 1mL of 0.5mM EDTA, then bringing to a final volume of 50mL of water. We used 400uL of this Xbuffer to resuspend the cell pellet of approximately 2.5e9 diploid stationary-phase yeast cells, transferred this into screw-top bead beating tube with ~0.3mL of glass beads, added ~20ug of RNAseA (Monarch NEB), and agitated using a bead beater Biospec 607 for 5min. After this, tubes were incubated at 65˚C in a waterbath for 30min. 400uL of 24:1 chloroform:isoamyl-alcohol was added to each tube, these were closed and inverted vigorously to mix, then put back at the 65˚C waterbath for 15min. Tubes were centrifuged for 2 minutes at room temperature at max speed in a benchtop centrifuge, then the supernatant was removed to a new eppendorf tube. To this, 600uL more 24:1 chloroform:isoamyl-alcohol was added, and vortexed to mix well before centrifuging for 2 minutes at max speed. The supernatant (translucent yellow at this stage) from this was carefully aspirated to a new tube. Approximately 0.7x volumes of isopropanol was added, inverted to mix, and spun at 2 minutes max speed. This pellet (large white, chalky) was washed with 1mL of 70% ethanol, then the supernatant was removed and the pellet dried for 10 minutes at room temperature. To this pellet, we added 200uL of TE (10mM Tris 1mM EDTA) and incubated on bench for 30min to dissolve with two bouts of vortexing during incubation. To this 200uL of resuspended pellet we added 1 volume (200uL) of the QX1 buffer from the Qiaex2 DNA extraction kit (Qiagen) [56]. This was immediately spun for 30 seconds at max speed, and the supernatant aspirated to a new tube. To this supernatant we added 30uL of the Qiaex2 DNA-binding solution, and this was vortexed to mix at 2 minute intervals across a 10 minute room temperature incubation. This was then spun 30 seconds at 16,000g and supernatant aspirated and discarded. The pellet was washed twice by completely resuspending it in 0.5mL of PE (Qiaex2 kit) then spinning again to pellet and discard supernatant. After this, the supernatant was completely removed and pellet dried 10-15min at room temperature. The pellet was resuspended in 30uL of EB (10mM Tris) and incubated at 50˚C for 10min, then spun 30s as before. Supernatant was aspirated to a new tube, then the pellet was again resuspended with 30uL of EB and incubated at 50˚C for 10 minutes.

This was spun again, the supernatant was pooled with the previous eluted supernatant, and the genomic DNA content was estimated with a Qubit fluorimetric Broad Range dsDNA kit. Yield was typically about 15ug by this metric, with a Nanodrop returning a 260/280nm ratio of about 1.9 and a 260/230 ratio of about 1.8–2.3.

## PCR of double-barcode amplicons for Illumina sequencing

To amplify the double-barcode loci for amplicon sequencing we devised a protocol that robustly amplifies from >1ug (qubit-estimated) of genomic dsDNA from the previous extraction step, with minimal UMI-primer carryover (S8 Fig). We setup 50uL PCR reactions with 1x OneTaq GC Buffer (NEB), 400uM of additional $MgSO_4$, 200uM each dNTP, 500nM each primer, 0.25uL of OneTaq polymerase (NEB), and 1ug of genomic DNA sample from the previous extraction step. For this first round we used primers as denoted in S6 File, S10 and S11 Tables, which amplify from the engineered priming sites to amplify each double-barcode locus while adding on a random barcode (UMI) and known barcode (sample index) outside of the amplified locus. These were run on a BioRad T100 thermocycler to melt at 94˚C for 4 minutes, before going through 4 cycles of 94˚C for 1 minute, 52˚C for 30 seconds, and 68˚C for 1minute. Eight 50uL reactions were run for each timepoint sampled in this work, and so the products of this first reaction were pooled together and ethanol precipitated to concentrate. These were resuspended in 25uL of water and immediately purified using 50uL of Ampure XP beads, washing twice with 80% freshly prepared ethanol. After drying, we added 20uL water, resuspended the beads, incubated for 5 minutes at room temperature, then added 11uL of a 20% PEG 8000 2.5M NaCl mixture and mixed the solution well with a pipette. After 5 minutes at room temperature, this was separated using a magnetic rack for 5 minutes, then the supernatant was removed to a fresh new tube. Presumably the large genomic DNA remain bound on the Ampure beads in this buffer, while the short amplicons are in the supernatant. So, the supernatant was ethanol precipitated, resuspended in 17uL distilled water, and quantified using a Qubit High-Sensitivity dsDNA assay. 15uL of the first round products were mixed into a second PCR reaction that contained 1x KAPA HiFi GC Buffer (Roche), 300uM dNTPs, 300uM each primer, and 0.02 U/uL KAPA HiFi DNA Polymerase (Roche) in a 30uL volume. For this round 2 PCR, we used sample multiplexing primers as denoted in S11 Table that amplify the double-barcode locus but add on the p5/p7 Illumina sequences as well as additional sample indices in the index-read position. This was run on a BioRad T100 thermocycler as 95˚C for 1 minute to melt, then 16 cycles of 98˚C for 20 seconds, 55˚C for 15 seconds, and 72˚C for 15 seconds. The products were gel purified (Zymo Gel Clean and Concentrator), then quantified using Qubit High-Sensitivity dsDNA fluorimeter kit before submission for sequencing on Illumina HighSeq runs at NovoGene (S8 Table).

## Illumina double-barcode amplicon-sequencing data analysis pipeline

This Illumina sequencing data was processed to generate per-lineage fitnesses for each selection. This pipeline was written in Nextflow and is available at the git repos linked at the start of Methods. Three lanes of Illumina HighSeq data were generated by NovoGene (S8 Table), with sample multiplexing using the primer indices as indicated above. We first used 'itermae' (https://gitlab.com/darachm/itermae/) to apply fuzzy regular expressions in robustly extracting the sample index, UMI, and lineage barcodes from each set of forward/reverse reads and filter on adequate read quality. Each different type of barcode on each read was clustered using 'starcode'(https://github.com/gui11aume/starcode) [111], with the lineage barcodes generated from the long-read barcode-annotation step being clustered along with the lineage barcodes from the Illumina data to correct any small systematic misannotations. Counts per sample for

each unique (distinct UMI) pair of lineage barcodes were analyzed to estimate chimera rates by fitting a model of how each sample and lineage barcode on either the forward or reverse read predicts the observation of these two specific sample and lineage barcodes together. We fit this model to lineage barcode combinations that we did not mate together (thus known chimeras), and found this to well fit the abundances of known chimera combinations ($R^2$ of 0.98, 0.94, and 0.95 for each replicate A, B, and C respectively). We then used this model to subtract the expected contribution of chimeras to each lineage barcode combinations, and the total chimera adjustment (1.8%, 4.9%, and 4.8% of for each replicate A, B, and C respectively) appears reasonable given the stated lower limit (~1.5%) on chimera rate using Illumina's ExAmp chemistry. These chimera-adjusted counts were then filtered for lineages that appear in at least three timepoints or are seen with at least five counts in only one timepoint. These counts are then subject to analysis with FitSeq [77], software that iteratively fits a model of Wrightian fitness to the counts data while modeling the effect of a rising population average fitness on each individual lineage's relative proportions (i.e. abundance counts). We adapted the reference python version of this tool (https://github.com/FangfeiLi05/PyFitSeq) [78] to elaborate better control of the model fitting process and user interface, as well as packaging on PyPi (https://github.com/darachm/fitseq, https://pypi.org/project/fitseq/) [112]. Various GNU tools, such as parallel [113], were instrumental to this pipeline. The resulting fitness per double-barcode lineage in each replicate were then analyzed in R.

## Analysis of lineage fitnesses into ORF-ORF PPI calls

Using R [114] we ran several scripts (available in git repo linked at start of Methods section). First we evaluated some quality-control metrics and filtered out double-barcode lineages that (1) did not fit a model by FitSeq with at least an $R^2$ of 0.5 or (2) had less than 5 counts in the initial timepoint sample. In the data that passed this filter, we excluded putative dropout lineages (as described in the text). For each group of lineage barcodes representing an ORF-ORF combination, we identified the lineages that were observed to be at least one standard deviation (of the distribution of all fitnesses for all lineages) away from the specific median fitness of this ORF-ORF combination. We removed these putative dropouts as long as at least three lineages remained, starting with the most extreme absolute deviation in fitness. This conservative heuristic removed some outlier lineages (S6 Fig), but there is room for further technical development to identify and mitigate these dropouts before measurement (Discussion). To facilitate interpretation we re-scaled each replicate's lineage fitnesses such that 0 was the median fitness of the NULL-F[1,2 [x NULL-F[3] double-negative controls and 1 was the median fitness of the NULL-F[1,2] x F[1,2]-F[3] covalent fusion positive control, however we believe this should not affect the calls of PPIs as each scaling and statistical test is performed within each replicate.

We used a t-test to detect groups of lineages with higher than expected fitnesses, but false-positives can arise if the group variance of fitnesses is very small by random sampling. To make the analysis more robust to this, we moderated the variance of each group of ORF-ORF combinations towards the variance for that set of NULL-F[1,2] x vORF-F[3] controls (S9 Fig). The concept is widely used in gene expression analysis [96]. We used this moderated variance approach to test if the fitnesses of each ORF-ORF combination group of lineages is higher than the fitnesses of the group of NULL-F[1,2] x vORF-F[3] control lineages. We then used the Benjamini and Hochberg method [115] on the distribution of calculated p-values to generate a False Discovery Rate, and called a hit in a replicate as an ORF-ORF combination with an FDR $< 0.05$. See the analysis scripts in the pipeline as linked from the beginning of Methods section.

## Detecting potential biological significance of PPI calls

To interpret the hits and shed light on how these might help generate biological hypotheses for future *in vivo* or *in vitro* assays, we analyzed the list of 74 PPIs where a hit had been detected in at least one of three replicates. Looking at the IMEX Coronavirus dataset provided by IntAct [80], we filtered this table for interactions between human and SARS-CoV-2 proteins of types "physical association", "direct interaction", "proximity", or "association", then used a hypergeometric test to determine how likely it was to that we called these 74 ORF-ORF combinations as a PPI, given that list of other putative PPIs. We then used the 'clusterProfiler'package [82] to test for significant enrichment of PROSITE domain annotations, annotations of "membrane" or "intramembrane" domains in Uniprot, Pathway annotations from Pathway Commons (v12, https://www.ncbi.nlm.nih.gov/pmc/articles/PMC3013659/), and GO terms obtained from Uniprot (GO version 2022-03-22). Reported p-values are all adjusted using the Benjamini and Hochberg method [115].

## Using REDIseq to efficiently parse the PPIseq haploid viral-ORF strain library

Cell pools of haploid yeast containing mDHFR tagged-ORF expression constructs were generated using methods described above. For the three pools of haploid yeast expressing ORFs similar to ORFs in the SARS-CoV-2 genome, these were inoculated and grown overnight in 5mL of YPD+Hygromycin. In the morning, these were collected by centrifugation and resuspended in the same volume of distilled sterile water, then diluted 1:100 in the same. These were counted by hemacytometer and adjusted to a density of approximately 5e5 cells per mL. Then, we used a Labcyte Echo 550 to dispense 7.5nL or 5nL into all wells of four 384-well Labcyte-approved plates containing 30uL of YPD+Hygromycin+Carbenicillin in each well. There were grown overnight at 30°C, and for two of the three pools these were agitated and OD was measured before re-innoculating into empty wells using LabCyte dispensing of 7.5nL or 5nL of the pre-cultured and diluted cell pool prepared similarly as above. pLSL53 was the exception to this, and instead we dispensed droplets into all wells under the logic of the established colony competitively excluding the potentially invasive new cell. All of these underwent four rounds of re-innoculation with dilute cell suspension and overnight culture at 30°C, before 30uL of 30% glycerol was added to each well. This resuspended array was pinned to YPD before the primary liquid culture (at 15% glycerol) was frozen at -80°C. A complementary array of iSeq2 alpha strains [52] was pinned out from frozen stocks, and the subsequent day the two arrays of MATa (vORF PPIseq strains) and MATalpha (iSeq2 interaction barcoders) were pinned together and grown overnight on YPD. These matings were pinned to SC-met-lys media to select diploids, then to YPGal to induce recombination of the barcodes and split-URA3 marker. The next day, arrays were pinned to SC-URA to select for these faithful double-ORF double-barcode diploids, and these resulting arrays were scraped into pools and DNA was extracted using the same procedure as described above for the pool-selection samples. PCR was performed similar to as described above, but with less input genomic DNA (200ng), more cycles during PCR (8 for first round and 20 for second round), and different primers for amplification of the iSeq2 side of the amplicon with appropriate sample indicies (S11 Table). Products were gel-purified (Zymo Gel Clean and Concentrate kit), quantified with Qubit, and pooled for sequencing (S8 Table). This sequencing was analyzed to determine colonies that appeared to be pure colonies of yeast containing particular known vORF barcodes. Colonies with barcodes annotated (above) as mapping to a clearly correct ORF were identified. With these results we identified colonies that comprise replicate sets of yeast vORF colonies. With this information we used a Singer PIXL colony picking robot to rearray the desired colonies to

liquid media in a 384-well plate with 75uL YPD+Hyg+Carb. After a day of growth, obviously empty wells were re-picked by the PIXL, and after another night of growth the collection was brought to 15% glycerol and frozen.

## Confirming hits of the PPiSeq screen

To verify that the method of scaling the established mDHFR PPI assay works with the scaling method of PPIseq, we repeated the assay with clonal arrays on agar. Colonies containing expression constructs of mDHFR-tagged vORFs were arrayed as previously described (REDIseq), so we picked colonies representing several vORFs that were participants in detected PPIs. We then also re-generated clonal strains expressing mDHFR-tagged hORFs. To do this, we used Gateway LR Clonase as per manufacturer's instructions to integrate the ORFs from entry vectors (picked from the human ORFeome collection v8.1) into a pre-barcoded pool of pSL51-derived plasmids, in vitro. These pools were each transformed into competent 10beta cells prepared with a Mix and Go kit (Zymo Research), selected with LB+Carb media, and clones were picked into a 96-well plate for screening. These were screened first for growth on LB+Chloramphenicol to eliminate colonies that retained the Gateway negative-selection cassette, then screened by PCR across the ORF (primers oSL269 and oSL26). Plasmids with the right ORF inserted were transformed and integrated into the landing pad in yeast strain ySL508 using a similar transformation procedure as previously described, and clones were again screened with colony PCR using oSL231 against oSL10 (for pSL51-derivatives) and oSL1160 against oSL450 (for pSL737-derivatives). Yeast colony PCR products were Sanger sequenced, and a clone of each ORF construct were picked into the margin columns and rows of a 96-well plate (7 vORFs and 11 hORFs, as in S19 Table). These were then crossed in the same multi-well plate in liquid YPD for several hours before being pinned to YPD+G418+Hyg agar media. Colonies were then pinned to YPGal, and the resulting colonies after growth were pinned to SD+his+leu to select for faithful double-ORF double-barcode recombinants and frozen at -80˚C with 15% glycerol. The same procedure was carried out for the controls, wherein the pooled libraries pLSL47 (F[1,2]-F[3] positive control), pLSL48 (F[1,2]-NULL negative control), and pLSL49 (NULL-F[3] negative control) were grown in liquid media overnight and then crossed against each other and the relevant complementary ORF assay strains—ie pLSL48 (F[1,2]-NULL) against all F[3]-tagged vORFs and pLSL49 (NULL-F[3]) against all F[1,2]-tagged hORF strains. The final mated diploid assay strains were grown overnight in SD+His+Leu, then back-diluted 1/20 into SD+His+Leu+1ug/mL MTX liquid media and grown until saturation, then pinned in quadruplicate to SD+His+Leu+MTX media with concentrations of 100, 10, 1, and 0 ug/mL MTX. These were grown two days, then scanned. The images were analyzed using a custom R script (see Code Access at beginning of Methods section) using EBImage to segment colony from background. filter out noise, and register colonies to positions before total colony size was used as a proxy for growth during this assay. The 100ug/mL MTX analysis was reported because it was the clearest effect and the effects appeared completely consistent between different MTX concentrations.

## Supporting information

**S1 File. A protocol for high-throughput library-scale yeast transformation.** (HTML)

**S2 File. A protocol for large-scale yeast genomic DNA extraction for barcode-sequencing.** (HTML)

**S3 File. A protocol for PPIseq barcode-sequencing PCR (mjtF—evoR2 priming sites version).**
(HTML)

**S4 File. pSL51 Genbank file.**
(GBK)

**S5 File. pSL737 Genbank file.**
(GBK)

**S6 File. A diagram to explain the nomenclature used for multiplexed sequencing primer tables.**
(TIF)

**S1 Fig. The modified PPIseq system still reproduces PPI detection.** Raw scans of plates used for initial verification that modified plasmids and modified strains still reproduce the mDHFR assay results. Layout as described in S13 Table.
(PDF)

**S2 Fig. Lineage traces of all PPI hits.** For each timepoint (x-axis), the relative frequency of each lineage (y-axis) is shown for each replicate and each ORF-ORF combination (panel facetting). 'Hit' or 'non-significant' label indicates if this data was called as a significant PPI hit, and line color indicates the lineage fitness.
(PDF)

**S3 Fig. Confirming PPI hits using a plate-growth assay.** Raw scans of some regenerated strains assaying PPIseq hits were grown using clonal growth on agar MTX plates (see Methods). Layout as described in S19 Table.
(TXT)

**S4 Fig. Technical consistency of lineage quantification with the first timepoint of the PPIseq assay.** PPIseq double-barcode counts are well correlated. Two biological samples of the initial timepoint's lineage pool were subject to independent DNA extraction, PCR, and sequencing. A) Chimera-adjusted counts from each sample (Methods). B) $R^2$ for a subset of the dataset (y-axis), calculated for all observations with both counts greater than the minimum counts threshold (x-axis).
(TIF)

**S5 Fig. Examples of "dropout" lineages within the PPIseq dataset.** Dropout lineages are consistently less fit and not explained by ORF mutations. A) Examples of lineages for tagged human DHFR and tagged vORF positive controls that are expected to have high fitness. Proportion of counts (y-axis) per lineage is shown for each timepoint (x-axis). Some lineages are not detected in some replicates. B) The un-scaled fitness of each lineage for all tagged human DHFR and tagged vORF positive control lineages are plotted for each replicate (x and y axes). Color indicates if the lineage contains an ORF that was annotated by the long-read plasmid annotation to contain a non-synonymous mutation.
(TIF)

**S6 Fig. Examples of outlier filtering applied to the PPIseq lineages in order to remove spurious "dropout" lineages.** Examples of outlier filtering. For each panel the un-scaled fitness of each lineage for all tagged human DHFR and a certain tagged vORF positive control lineages are plotted, for replicate A (x-axis) against replicate B (y-axis). Color indicates if the lineage

passed the outlier filer or not (FALSE indicates exclusion).
(TIF)

**S7 Fig. Dotplots of the fitnesses of vORF-DHFR positive-control strains to show the effect of NSP2 on growth.** NSP2 tends to have a slight fitness advantage with human DHFR positive control. For each vORF (x-axis) for each replicate (color), the scaled fitness of each lineage of tagged human DHFR and tagged vORF lineages are plotted (y-axis), and the mean of each grouping is indicated by the solid line.
(TIF)

**S8 Fig. Confirming the efficiency of the new barcode-sequencing priming sites using qPCR.** New PPIseq priming sites and PCR protocol are efficient and effective at generating UMI-containing libraries. **A)** qPCR of Round 1 PCR (genomic DNA template and genomic priming sites) and Round 2 PCR (purified Round 1 template with p5/p7-containing primers) were set up with Kapa HiFi polymerase and SybrGreen dye (Methods). Plotted is the "relative concentration" of the template as estimated from $2^{-C_t}$ where $C_t$ is the maximum second-derivative cycle threshold calculated by the AB HT7900 qPCR machine. Dilution is from a serial dilution of the template. **B)** A similar experiment was performed, but the Round 2 PCR was carried out with template either from a normal Round 1 PCR (primers added, thermocycled, then template purified), or from a reaction where the primers were withheld but a purified product of a previous Round 1 template was added, or from a reaction where the primers were withheld until after the cycling but primers and a purified Round 1 template were added just before purification. Three different purification strategies (raw with no purification, column purification, or ExoI treatment) show different efficacy at repressing the primer-after reaction to the same level as the no-primer reaction.
(TIF)

**S9 Fig. Showing trends in variance shrinkage used to moderate the effects of few-lineage underestimates of lineage-group variance.** A summary of the variance shrinking approach to moderate false positives. A), B), C) show for each replicate the observed variance of fitnesses within an ORF-ORF lineage group (x-axis) and the moderated, or shrunk group fitness variance used for the one-sided t-test. D) The distribution (y-axis) of the variance in fitnesses per ORF-ORF group (x-axis) is shown for the observed and moderated variance (linetype) for each replicate (color).
(TIF)

**S10 Fig. The number of barcoded lineage clones found by REDI-seq, for each tagged vORF in the clone isolated.** REDIseq is a viable strategy for converting pooled PPIseq haploid precursor yeast strain pools into arrayed libraries. We used dilution to separate pools of PPIseq haploids (yLSL52, yLSL53, yLSL54, yeast that contained integrated F[3]-tagged vORFs) 9 384-well plates. Many wells were left empty by this random dispersion process, but the entire plates were subject to REDIseq wherein these were crossed to plates of compatible iSeq 2.0 strains with known barcodes. For each tagged vORF (x-axis), we recovered some number of uniquely-barcoded haploid clones (y-axis).
(TIF)

**S11 Fig. The distribution of the number of lineages per PPI tested in each replicate.** The number of lineages per ORF-ORF combination is well dispersed. The distribution (y-axis) of the number of double-barcoded lineages (x-axis) that assay each ORF-ORF pair is shown. Linetype indicates different replicates.
(TIF)

**S12 Fig. No-nitrogen recovery appears to increase yeast LiAc transformation efficiency.**
(PDF)

**S1 Table. Plasmids and pools of plasmids used in this work.**
(CSV)

**S2 Table. SARS-CoV-2 pooling strategy used in this work.**
(CSV)

**S3 Table. Yeast strains, pools, and arrays used in this work.**
(CSV)

**S4 Table. vORF barcode annotations used in this work.** This table shows which vORF (viral ORF) a particular barcode corresponds to.
(CSV)

**S5 Table. hORF barcode annotations used in this work.** This table shows which hORF (human ORF) a particular barcode corresponds to.
(CSV)

**S6 Table. Lineage double-barcodes of control strains used in this work.**
(CSV)

**S7 Table. Lineage annotation table used in this work.** This table shows which pair of barcodes corresponds to which lineage tracked in the double-barcode sequencing.
(TXT)

**S8 Table. Sequencing datasets used in this work.**
(CSV)

**S9 Table. Primers used for strain engineering in this work.**
(CSV)

**S10 Table. Sample multiplexing primer usage for the PPIseq screen.** This table denotes the sequence of the sample multiplexing primers used for the PPIseq screen, the main bulk of the paper (as shown in main second figure). Nomenclature corresponds to as in S6 File Diagram.
(CSV)

**S11 Table. Sample multiplexing primer usage for REDIseq.** This table denotes the sequence of the sample multiplexing primers used for the REDIseq (isolation of the single vORF haploid strains). Nomenclature corresponds to as in S6 File Diagram.
(CSV)

**S12 Table. Sequences of primers used for preparing sequencing libraries.**
(CSV)

**S13 Table. Layout of strains on the plate for initial verification.** The plate layout for the experiment to initially verify that modified plasmids and modified strains still reproduce the same mDHFR assay results (S1 Fig).
(CSV)

**S14 Table. Sources of human ORFs.** The sources of each human ORF used for initial verification that modified plasmids and modified strains still reproduce the mDHFR assay results.
(CSV)

**S15 Table. Previously known PPIs.** A table of the PPIs amongst the ORFs used for initial verification that were previously reported in IntAct.
(CSV)

**S16 Table. Mean fitnesses of the population as estimated by FitSeq.**
(CSV)

**S17 Table. Lineage fitness table.** A table of the fitness of each double-barcoded lineage in each replicate, essentially our best estimate of the PPIseq signal for that lineage in that replicate of selection.
(TXT)

**S18 Table. PPIseq results.** A table of, for each combination of ORFs, if each combination was detected as a PPI, the FDR calculated, variances, effect sizes, lineages, mutation flags, and other features described in the table header.
(CSV)

**S19 Table. Plate layout for re-testing the PPIseq hits using clonal growth on agar MTX plates.** Layout of strains in the plate regenerated to confirm PPI hits using a plate-growth assay. See S3 Fig.
(CSV)

## Acknowledgments

We thank the members of Sasha Levy's lab at SLAC at Stanford, Gavin Sherlock's lab at Stanford Genetics, and the Steinmetz and Davis groups at the Stanford Genome Technology Center. In particular we thank Zhimin Liu, FangFei Li, Fabiana Gnatta, David Catoe, Michi Henri Tai, Angela Chu, Joe Horecka, Kevin Roy, Katja Schwartz, Weiyi Li, and Takeshi Matsui. Thanks to the University of Arizona Arizona Genomics Institute (AGI) core sequencing team for the paid PacBio sequencing service. We thank Fredrick Roth's group and others for quickly generating and freely sharing their collection of SARS-CoV-2 sequences, as well as the ORFeome Collaboration for generating the human ORFeome collection. The BioIcon of a genome sequencer in Fig 2 panel A is title "genomesequencer-2" and is by DBCLS, licensed under CC-BY 4.0 Unported https://creativecommons.org/licenses/by/4.0/. We thank the American people for the continued investment, by means of the National Institutes of Health, that enabled the training and sharing of the body of knowledge, techniques, and vision expressed here.

## Author Contributions

**Conceptualization:** Darach Miller, Adam Dziulko, Sasha Levy.

**Data curation:** Darach Miller.

**Formal analysis:** Darach Miller.

**Funding acquisition:** Sasha Levy.

**Investigation:** Darach Miller, Sasha Levy.

**Methodology:** Darach Miller, Adam Dziulko.

**Project administration:** Sasha Levy.

**Resources:** Darach Miller, Adam Dziulko.

**Software:** Darach Miller.

**Supervision:** Sasha Levy.

**Validation:** Darach Miller.

**Visualization:** Darach Miller, Sasha Levy.

**Writing – original draft:** Darach Miller, Sasha Levy.

**Writing – review & editing:** Darach Miller, Adam Dziulko, Sasha Levy.

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
