## [Decision Letter · Decision Letter 0]

24 May 2024

PONE-D-24-05663Pooled PPIseq: screening the SARS-CoV-2 and human interface with a scalable multiplexed protein-protein interaction assay platformPLOS ONE

Dear Dr. Miller,

Thank you for submitting your manuscript to PLOS ONE. After careful consideration, we feel that it has merit but does not fully meet PLOS ONE’s publication criteria as it currently stands. Therefore, we invite you to submit a revised version of the manuscript that addresses the points raised during the review process.

We look forward to receiving your revised manuscript.

Kind regards,

Hin Fung Tsang

Academic Editor

PLOS ONE

Journal Requirements:

 [This work was solely funded by the National Institutes of Health National Institute for Allergies and Infections Diseases (https://www.niaid.nih.gov/) grant R01 AI164530 awarded to Dr. Sasha Levy (SL) and carried out within Dr. Sasha Levy's research group at Stanford University. ].  

5. We notice that your supplementary tables are included in the manuscript file. Please remove them and upload them with the file type 'Supporting Information'. Please ensure that each Supporting Information file has a legend listed in the manuscript after the references list.

5. We notice that your supplementary figures are uploaded with the file type 'Figure'. Please amend the file type to 'Supporting Information'. Please ensure that each Supporting Information file has a legend listed in the manuscript after the references list.

6. We are unable to open your Supporting Information file [s4.gbk and s5.gbk]. Please kindly revise as necessary and re-upload.

8. We note that Figure s37 and s20 in your submission contain copyrighted images. All PLOS content is published under the Creative Commons Attribution License (CC BY 4.0), which means that the manuscript, images, and Supporting Information files will be freely available online, and any third party is permitted to access, download, copy, distribute, and use these materials in any way, even commercially, with proper attribution. For more information, see our copyright guidelines: http://journals.plos.org/plosone/s/licenses-and-copyright.

a. You may seek permission from the original copyright holder of Figure s37 and s20 to publish the content specifically under the CC BY 4.0 license. 

Reviewers' comments:

Reviewer's Responses to Questions

**Comments to the Author**

1. Is the manuscript technically sound, and do the data support the conclusions?

Reviewer #1: Yes

Reviewer #2: Yes

2. Has the statistical analysis been performed appropriately and rigorously? 

Reviewer #1: Yes

Reviewer #2: Yes

3. Have the authors made all data underlying the findings in their manuscript fully available?

Reviewer #1: Yes

Reviewer #2: Yes

4. Is the manuscript presented in an intelligible fashion and written in standard English?

Reviewer #1: Yes

Reviewer #2: Yes

5. Review Comments to the Author

Reviewer #1: The manuscript by Miller et al. presents the application of a sequencing-based method for identifying protein-protein interactions (PPiSeq), a technique originally pioneered by the Levy lab in 2017. This study leverages PPiSeq to delineate viral-host protein-protein interactions (PPIs), a research area of significant interest that is in need of innovative scalable methodologies. By determining the host and viral partner proteins, this type of work can offer valuable insights into viral replication processes and facilitate the development of novel antiviral therapies. The experimental design and analyses conducted are robust, rendering the manuscript suitable for publication in PLoS One. While no major revisions are required, a few minor adjustments are recommended prior to acceptance.

Major Revisions:

None

Minor Revisions:

1. Line 130: Correct “lading” to “landing”

2. The authors conduct a statistical analysis to demonstrate that their method significantly overlaps with datasets previously published by IMex. However, despite the statistical validation, the overall number of overlaps remains small. It would be beneficial for the authors to delve deeper into some of the novel protein-protein interactions (PPIs) that have not been identified before. The authors should provide some discussion of why this approach might have picked up interactions the previous approaches missed. On the other hand, although more challenging, it would also be insightful to investigate which PPIs, potentially replicated by different laboratories independently, do not appear among the 74 hits identified through PPiSeq. Expanding the discussion to address these aspects would greatly enrich the manuscript, offering a more comprehensive understanding of the method's significance and the novelty of the findings.

3. Within the 113 top hits (FDR<0.05) we see NSP2 52 times, ORF7 22 times, NSP4 16 times, NSP5C145A 13 times – can the authors determine if there is any relationship between the expression levels of these viral genes and observing interactions?

Reviewer #2: Minor comments, mainly editorial. Some comments on statistical reporting (format of p values) and complexity of figures, which could / should be simplified if possible. Temperature should also include °C and not only be reported as C in the methods section.

See attached for more information.

6. PLOS authors have the option to publish the peer review history of their article (what does this mean?). If published, this will include your full peer review and any attached files.

Reviewer #1: No

Reviewer #2: No

---

## [Author Response · Author response to Decision Letter 0]

14 Aug 2024

Please see the Response to Reviewers document uploaded. For completeness, here I am copy and pasting that document below:

Date: 2024-06-10

To: PLoS One

Subject: Response to Reviewers

PONE-D-24-05663R1

In this document we would like to reply to the generous feedback provided by the reviewers. I will reprint their comments in red, and reply inline using black text. Because the EM system does not permit text formatting, I will also prepend all Reviewer Comments using the characters “> “

> Reviewer #1: The manuscript by Miller et al. presents the application of a sequencing-based 

> method for identifying protein-protein interactions (PPiSeq), a technique originally pioneered 

> by the Levy lab in 2017. This study leverages PPiSeq to delineate viral-host protein-protein 

> interactions (PPIs), a research area of significant interest that is in need of innovative 

> scalable methodologies. By determining the host and viral partner proteins, this type of work can

> offer valuable insights into viral replication processes and facilitate the development of novel antiviral 

> therapies. The experimental design and analyses conducted are robust, rendering the manuscript 

> suitable for publication in PLoS One. While no major revisions are required, a few minor adjustments 

> are recommended prior to acceptance.

We thank Reviewer 1 for the thoughtful discussion, and we appreciate the reviewer’s comments to allow us to reconsider these points and update the manuscript accordingly.

> Major Revisions:

> None

> 

> Minor Revisions:

> 1. Line 130: Correct “lading” to “landing”

We corrected this typo.

> 2. The authors conduct a statistical analysis to demonstrate that their method significantly overlaps 

> with datasets previously published by IMex. However, despite the statistical validation, the overall 

> number of overlaps remains small. It would be beneficial for the authors to delve deeper into some 

> of the novel protein-protein interactions (PPIs) that have not been identified before. The authors 

> should provide some discussion of why this approach might have picked up interactions the 

> previous approaches missed. On the other hand, although more challenging, it would also be 

> insightful to investigate which PPIs, potentially replicated by different laboratories independently, do 

> not appear among the 74 hits identified through PPiSeq. Expanding the discussion to address these 

> aspects would greatly enrich the manuscript, offering a more comprehensive understanding of the 

> method's significance and the novelty of the findings.

The reviewer is correct that we only re-discover a small fraction of the PPIs previously reported among the set of proteins considered here. However, this low overlap is typical for PPI screens (or even within screens, as seen in the most recent work by Fritz Roth and colleagues), and so we expanded the discussion in the limitations section to again highlight the 6 / 74 overlap, specifically by adding the following sentences to the paragraph discussing limitations, starting around line 433:

"The seemingly low overlap with previously reported PPIs (6 of 74) is consistent with the difficulty of reproducible detection while limiting false-positives in the massive search space of PPI screens [89], but this also likely reflects that the mDHFR split-tag suffers from the same kinds of limitation and bias as other assay modalities. While the mDHFR split-tag reporter may be more sensitive to membrane-associated proteins it is also likely biased against detecting PPIs of nuclear proteins [17], for example."

Additionally, in the line 438 originally reading "the consistency with previous SARS-CoV-2-human screens", we replaced "consistency" with "significant overlap". We have also previously directed attention to a key difference of the mDHFR-split-tag system in the first paragraph of the Discussion that remains unchanged, specifically:

"PPIseq uses the mDHFR split-tag as our PPI selection system, which has high sensitivity to transmembrane protein PPIs and serves to complement previous SARS-CoV-2-human PPI screens."

The second part of Reviewer 1’s suggestions appear to suggest a systematic analysis of the interaction of protein features with PPI detection modalities. This is indeed a useful and rewarding study to propose, and this kind of work has been the subject of past papers in this field. This would also be interesting to repeat across viral-human proteins and indeed some previous work has done specifically that (Halehalli and Nagarajaram 2015 among others). However, that analysis would be best performed in analyzing across many viral-host protein examples across the field, and would be a large undertaking that we believe is greatly out of the scope of this paper and beyond our immediate expertise.

> 3. Within the 113 top hits (FDR<0.05) we see NSP2 52 times, ORF7 22 times, NSP4 16 times, 

> NSP5C145A 13 times – can the authors determine if there is any relationship between the 

> expression levels of these viral genes and observing interactions?

The reviewer also asked "can the authors determine if there is any relationship between the expression levels of these viral genes and observing interactions?" To address this we first modified the discussion to highlight that we cannot determine potential differences of expression between ORFs by adding a phrase near line 427:

 "We have not verified that each of the thousands of ORFs integrated into the tag-fusion construct are faithfully and accurately expressed in each strain, that the expression of each tagged-ORF is similar, or that they fold to generate all functional domains relevant for an interaction."

However, the question can also be interpreted as asking if there is an interaction between the observation of a PPI here and the in vivo expression level, ie the expression during an infection of a mammalian cell. This is an interesting question in light of observations made by Allan Drummond, Emanuel Levy, David Enard, and many others over the years, where a hypothesis could be made that a natural history of high-expression during infection could be associated with selection against potentially deleterious interactions with cellular proteins. To test this, we downloaded a proteomics dataset from this paper (https://www.ncbi.nlm.nih.gov/pmc/articles/PMC8220945/ , last author Kutluay) that reported the abundance of proteins upon a time-course of SARS-CoV-2 infection of H522 cells. There are only four ORFs that are both quantified in this dataset as well as detected with at least one PPI in our work, and so we can compare the number of PPIs detected against the protein expression levels. If we do so for each timepoint, I do not see a significant correlation (p-value < 0.05) - but all correlations are negative, consistent with the hypothesis. If we pool all timepoints (and do not adjust for this multiple testing!), then R calculates a p-value of 0.02859, with a negative correlation between abundance and PPI hits. Thus, the reviewer poses an intriguing question but we hesitate to report this preliminary analysis given the scarce observations considered here (only four proteins "N", "NSP2", "ORF7A", "NSP4"). But we believe that our dataset can be part of an analysis for future researchers to compare across assay modalities, tissues, proteins, and viruses.

> Reviewer #2

We thank Reviewer 2 for the kind words, thoughtful consideration, and detailed attention - this diligence ensures that scientific literature remains clear and widely accessible both now and into the future.

> My assessment / comments are as follows:

 1. > Is the manuscript technically sound, and do the data support the conclusions?

> Yes

 2. > Has the statistical analysis been performed appropriately and rigorously?

> Yes

> I noticed that the format for reporting p values differ throughout the manuscript (p value < 1e5 > [line 329], p value < 0.05 [line 357], p value < 1e-6 [line 337], etc.). Is there a reason?

This is an important point. In literature p-values can be reported as the exactly calculated value or as being less than a threshold, and in this work we attempted to compromise between the two by reporting the lowest threshold they were under. Upon considering the Reviewer's comment we realized that instead the style we used just shares the negative aspects of each style. So to fix this, we have changed all p-values and FDR below this typical 0.05 threshold to just reference the threshold, for example as "p-value < 0.05". We believe this fixes the issue with the usual correct notation of p-values as being below a commonly accepted threshold of 0.05.

 3. > Have the authors made all data underlying the findings in their manuscript fully available?

> Yes

 4. > Is the manuscript presented in an intelligible fashion and written in standard English?

> Yes

 5. > Review comments to the Author:

> Mostly editorial apart from above comments.

 1. > Pag 7 – inconsistent use of bold format

Corrected!

 2. > Page 10 – starting a sentence with a number (line 264) makes reading difficult. 

> The sentence needs to be reconstructed or 211,688 should be written out in full.

Corrected!

 3. > Page 14 line 337 – seems like a misplaced )

Corrected!

 4. > Page 16 line 410 – no space between “…s[81]”

Corrected!

 5. > Methods – unless this is customary for this specific journal, the degree symbol 

> must be used for temperatures (72°C instead of 72C)

We missed that we had left all temperatures written without a degree symbol, and so have added that degree symbol to correct all of these mistakes to adopt proper units notation.

 6. > The figures are quite complex and difficult to follow. For example, Figure 1 consists 

> of 10 sections – can this not be split into more, simpler and easier to follow figures?

We agree they are complex, but we do not see a way to reorganize the structure of the figures, and by extension the whole paper, without expending a great deal of effort for authors, reviewers, and editors. We believe, and hope the reviewers and editors will agree, that interested readers will be able to pull out the information they need from the paper. But we agree with the reviewer's comments as we have also found this structure to be difficult to work with as the paper grew. We will keep this experience in mind when designing how we begin our next publication.

 7. > Minor spelling mistakes.

We've used a spell-checker and have re-read the manuscript to identify any misspellings. Please bring to our attention any that we missed.

> I found this manuscript of exceptionally high quality and although it is fairly lengthy, I am of 

> the opinion that the information contained therein, justifies the length.

Thanks!

To the editor: In the course of making these edits, we also removed "molecular" on line 412 as that adjective seems inappropriate in that discussion. Please advise if we should return it.

Darach Miller

---

## [Editor Report · Decision Letter 1]

26 Aug 2024

Pooled PPIseq: screening the SARS-CoV-2 and human interface with a scalable multiplexed protein-protein interaction assay platform

PONE-D-24-05663R1

Dear Dr. Miller,

We’re pleased to inform you that your manuscript has been judged scientifically suitable for publication and will be formally accepted for publication once it meets all outstanding technical requirements.

Kind regards,

Hin Fung Tsang

Academic Editor

PLOS ONE
---

## [Editor Report · Acceptance letter]

6 Nov 2024

PONE-D-24-05663R1 

PLOS ONE

Dear Dr. Miller, 

I'm pleased to inform you that your manuscript has been deemed suitable for publication in PLOS ONE. Congratulations! Your manuscript is now being handed over to our production team.

Kind regards, 

on behalf of

Dr. Hin Fung Tsang 

Academic Editor

PLOS ONE